# Alignment of single-cell trajectory trees with CAPITAL

Reiichi Sugihara [1], Yuki Kato [1,2] ✉, Tomoya Mori [3] & Yukio Kawahara [1,2]

Global alignment of complex pseudotime trajectories between different single-cell RNA-seq datasets is challenging, as existing tools mainly focus on linear alignment of single-cell trajectories. Here we present CAPITAL (comparative analysis of pseudotime trajectory inference with tree alignment), a method for comparing single-cell trajectories with tree alignment whereby branching trajectories can be automatically compared. Computational tests on synthetic datasets and authentic bone marrow cells datasets indicate that CAPITAL has achieved accurate and robust alignments of trajectory trees, revealing various gene expression dynamics including gene–gene correlation conservation between different species.

Single-cell RNA sequencing (scRNA-seq) has enabled us to scrutinize the gene expression of dynamic cellular processes such as differentiation, reprogramming, and cell death. Since tracing a gene expression level of the same cell over some period of time is infeasible, *pseudotime* analysis with scRNA-seq snapshot data on a cell population from a tissue or an organ is of great value to obtain an approximate landscape of gene expression dynamics in those biological systems.

To model dynamic developmental processes for a given scRNA-seq dataset, various computational tools were developed to predict a *cell-state transition trajectory* or a *pseudotime trajectory*[1–5]. The predictable topology of trajectories depends on the method of choice, ranging from a simple linear structure to a tree with branches, and even a complex cycle[6]. A comprehensive comparison among existing trajectory inference tools was also provided to make an exhaustive investigation into their performance[7].

Comparison of pseudotime trajectories will provide a key to unveiling regulators that determine cell fates. For instance, trajectory comparison is used to investigate differences in gene expression dynamics between species (e.g. human vs. mouse), which will unravel evolutionary conservation and difference in the determination of cell fates such as regulation timing for an orthologous gene.

Recently, computational methods for aligning pseudotime trajectories across different datasets have been proposed[8,9]. They aim to align two single lineages across datasets with dynamic time warping[10], which is an analogy to a classical problem of pairwise sequence alignment, but differs in that multiple cells in one dataset may be matched with one cell in the counterpart at a time point. It remains, however, an unsettled question how one should select a pair of single lineages to be compared if the pseudotime trajectories include branches in each dataset, as the above methods based on dynamic time warping can deal only with linear trajectories for comparison. Furthermore, selecting a pair of trajectories to be compared requires accurate downstream analysis of a single-cell dataset and prior knowledge of a developmental pathway. Although one can consider inferring a common trajectory after integrating different scRNA-seq datasets[11–13], trajectory inference in this case could yield unexpected results since there is a possibility of merging cells that are not closely related.

In this work, we present a computational method for comparative analysis of pseudotime trajectory inference with tree alignment (CAPITAL). The aim of this work is to provide a method for aligning different scRNA-seq datasets even if their pseudotime trajectories include branches, so that one does not need to select linear paths in the trajectories to be compared beforehand. In the proposed algorithm, when a pair of different but related scRNA-seq datasets is given, CAPITAL seeks to infer a pseudotime trajectory that can include multiple branches for each dataset, and then to compute an optimal alignment between the two trajectories. Thorough computational tests on CAPITAL with synthetic scRNA-seq datasets in comparison with several state-of-the-art methods for data integration indicate that

[1]Department of RNA Biology and Neuroscience, Graduate School of Medicine, Osaka University, 2-2 Yamada-oka, Suita, Osaka 565-0871, Japan. [2]Integrated Frontier Research for Medical Science Division, and RNA Frontier Science Division, Institute for Open and Transdisciplinary Research Initiatives (OTRI), Osaka University, 2-2 Yamada-oka, Suita, Osaka 565-0871, Japan. [3]Bioinformatics Center, Institute for Chemical Research, Kyoto University, Gokasho, Uji, Kyoto 611-0011, Japan. ✉e-mail: ykato@rna.med.osaka-u.ac.jp

CAPITAL has achieved accurate and robust alignments of trajectory trees. Next, a test with public scRNA-seq datasets of human bone marrow cells shows that CAPITAL was able to detect correct linear trajectories to be compared without manual selection. A further test with human and mouse bone marrow cells datasets tells us that CAPITAL was able to reveal not only similar expression patterns that seemed to be conserved but different molecular patterns between human and mouse, which would provide a key to unraveling novel regulators that determine cell fates.

## Results

### Overview of CAPITAL

The CAPITAL algorithm consists of the following two consecutive steps (cluster-based tree alignment and cell-based linear alignment):

1. Taking a pair of expression count matrices from two different scRNA-seq datasets as input, these matrices are preprocessed to construct their respective nearest neighbor graphs by using principal components derived from highly variable genes (Fig. 1a). The neighbor graphs are then used to identify communities (clusters) with the Leiden algorithm[14], and the representative cell in each cluster is defined as a virtual cell with the median measured in the reduced dimensional space, which is called a *centroid*. Respective trajectories are computed by finding the minimum spanning trees in the centroid-based graphs. Tree alignment of the centroids across different datasets is performed by a dynamic programming algorithm ("Methods").

2. Choosing a pair of aligned single lineages comprising the centroids according to the resulting tree alignment, the centroids in each dataset are decomposed into their original single cells. Next, each of the starting single cells in the aligned root cluster (e.g. A

and A' in Fig. 1b) is determined in a way that it has the longest distance to a cell among all other cells in the corresponding dataset. Computing an accumulated transition matrix for the cells along each path with the starting cell specified generates a pseudotime order[2]. Linear alignment of these single cells is then performed by a dynamic time warping algorithm for a set of genes of interest.

CAPITAL has an assumption that it can deal with a pair of single-cell trajectories with any number of branches (i.e. trees), but not cycles and disconnected graphs. Note that any pair of input trajectory trees can be accepted regardless of how similar they are in theory, but the comparison of datasets from completely different cell populations that are expected to have trajectories with different shapes is not relevant in practice.

### Benchmarking trajectory alignments with synthetic datasets

We first investigated how the clustering affects the ability of CAPITAL to align our synthetic datasets in terms of (i) the topology of the trajectories to be compared; (ii) the difference in the number of clusters across two datasets; and (iii) the total number of clusters in two datasets. The synthetic datasets used in these tests were generated by dyngen[15] with the backbone as a binary tree with three branches, which consist of 68 datasets of single-cell expression counts to be extended to more datasets for benchmarking alignments ("Methods"). In the first test to show the effect of global topological similarity between two datasets, the algorithm worked from the viewpoint of the normalized alignment distance ("Methods") even when minor changes were made to one of the trajectory trees to be aligned (e.g. removing a leaf, internal node, etc) (Fig. 2a). In contrast, a breaking example of

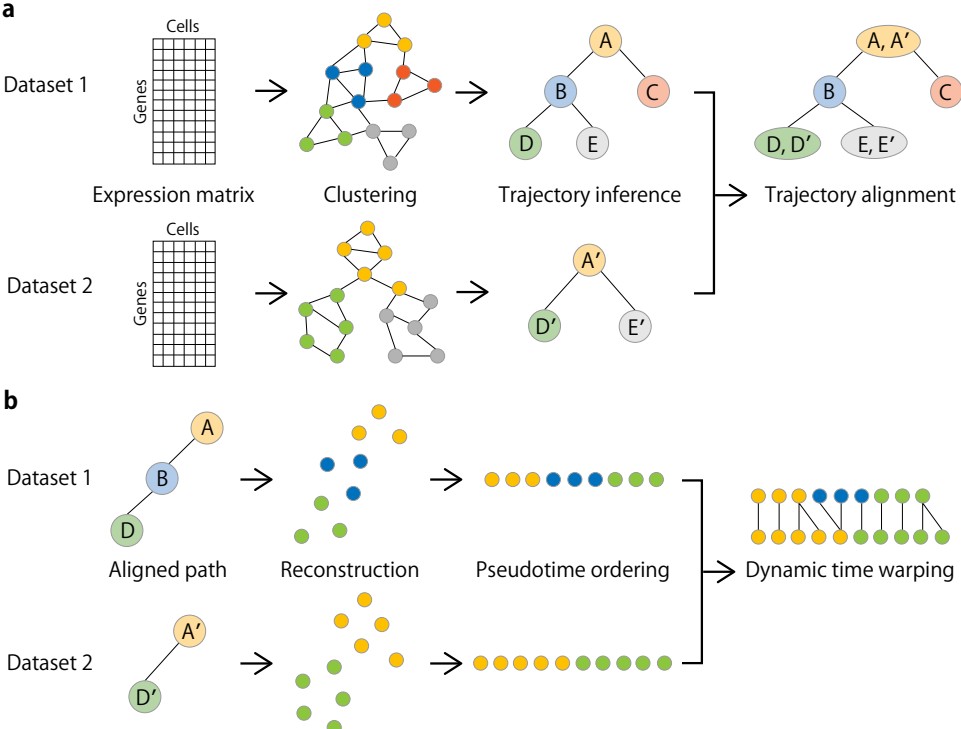

**Fig. 1 | Overview of CAPITAL algorithm. a** From an expression matrix of each dataset, a nearest neighbor graph is constructed where cells and their expression-based similarities are represented as vertices and weighted edges, respectively, and then clustered with community detection. Next, a centroid in each cluster is calculated to infer a trajectory of clusters by computing the minimum spanning tree. Finally, a tree alignment of trajectories of clusters is obtained by aligning one minimum spanning tree with the other across different datasets on the basis of a dynamic programming algorithm ("Methods"). **b** When a pair of aligned paths from the trajectory cluster alignment is chosen, all single cells are recovered with each labeled by the corresponding cluster, and ordered by diffusion pseudotime in each dataset. Linear alignment with dynamic time warping is then performed for a set of genes to investigate the dynamic relationship among single cells for those genes along the pseudotime.

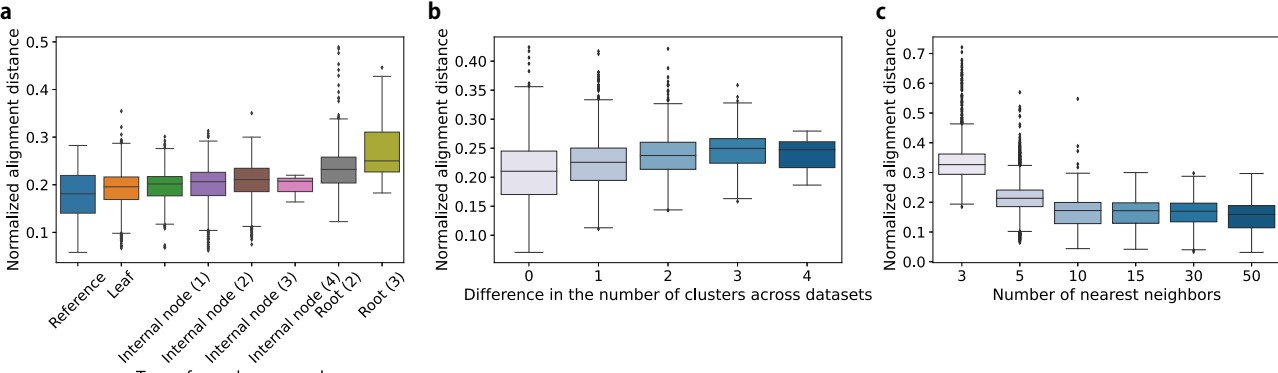

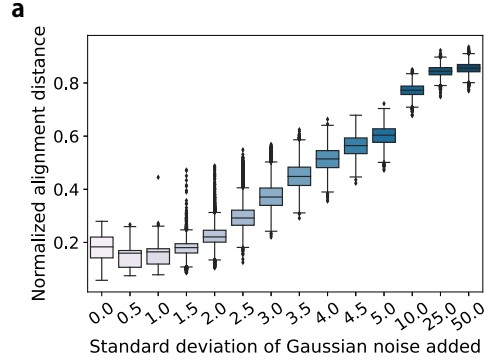

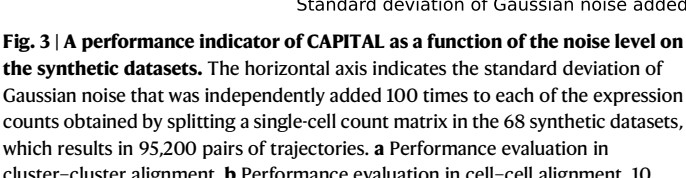

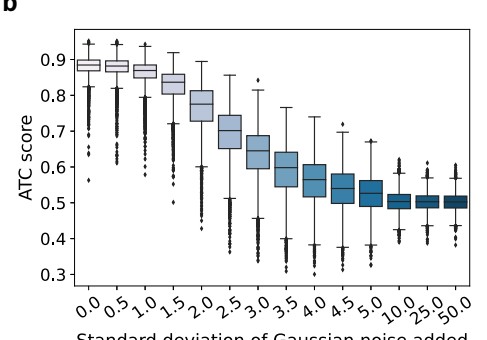

**Fig. 2 | Assessment of CAPITAL's feasibility in clustering quality on the synthetic datasets. a** A test in global topological similarity on 53,987 pairs of trajectories. Reference shows the results of trajectory alignments with no nodes in one trajectory tree removed. The other cases where a node was removed in one trajectory tree are classified into seven types. A number shown in parentheses on the horizontal axis shows the number of children of the node to be removed. 10 nearest neighbors were considered to build the nearest neighbor graphs. **b** A test in clustering similarity on 6800 pairs of trajectories. The number of nearest neighbors used is five. **c** A test in clustering resolution on 40,800 pairs of trajectories. The box plots show the median with bounds for the first and the third quartiles, and the whiskers indicate the minimum and the maximum that exclude outliers represented by points. Source data are provided in the Source Data file.

**Fig. 3 | A performance indicator of CAPITAL as a function of the noise level on the synthetic datasets.** The horizontal axis indicates the standard deviation of Gaussian noise that was independently added 100 times to each of the expression counts obtained by splitting a single-cell count matrix in the 68 synthetic datasets, which results in 95,200 pairs of trajectories. **a** Performance evaluation in cluster–cluster alignment. **b** Performance evaluation in cell–cell alignment. 10 nearest neighbors were considered to build the nearest neighbor graphs in all the tests. The box plots show the median with bounds for the first and the third quartiles, and the whiskers indicate the minimum and the maximum that exclude outliers represented by points. Source data are provided in the Source Data file. ATC average trajectory conservation.

removing the root with two or three children, which destroys a global topological similarity between two trajectories, shows that the normalized alignment distance clearly increased (the alignment performance declined). Next, the normalized alignment distance tended to increase gradually as the clustering dissimilarity increased, indicating that the clustering similarity affected the performance of the subsequent trajectory alignment (Fig. 2b). The third test in clustering resolution tells us that the number of nearest neighbors used to construct a trajectory has to be carefully chosen (Fig. 2c). Note that considering the larger number of nearest neighbors was likely to yield the smaller number of clusters (i.e. lower resolution), namely the alignment of trajectories with the larger number of clusters was more difficult than the opposite case. Taken together, suitable clustering for building a trajectory per dataset including choice of the number of nearest neighbors is necessary to enhance alignment performance of CAPITAL.

Second, we tested the robustness of CAPITAL measured by alignment accuracy on the synthetic datasets with increasing data noise ("Methods"). More precisely, we evaluated the alignment accuracy from two measures: (i) the normalized alignment distance for assessing the performance of cluster–cluster alignment; and (ii) the average trajectory conservation (ATC) score at the single-cell level

("Methods"). The rate of change in the normalized alignment distance was higher for the noise level of at least 2.0 than at most 1.5 (Fig. 3a), and that in the ATC score was higher for the noise level of at least 1.5 than at most 1.0 (Fig. 3b). Asymptotically, the two metrics deteriorated for the noise level of 3.0 or higher, as the corresponding cell space began to be shattered (Supplementary Fig. 1). Note that the ATC score of around 0.5 means that a true simulation time and a predicted pseudotime are most likely to have no correlation. Given that the noise of standard deviation of around 3.0 is unlikely to emanate from a typical dataset (e.g. standard deviation 3.0 was much larger than average 0.53 of the non-zero elements in the noise-free count matrices in our simulation), these results suggest that CAPITAL was robust to data noise to a certain degree at both the cluster-matching level and the single-cell alignment level.

Finally, to compare CAPITAL with three state-of-the-art methods of data integration in alignment performance, we ran Scanorama[12], scVI[11], and Seurat[13] on the combinations of the synthetic datasets to merge two respective datasets and perform common trajectory inference on that merged dataset. Note that pseudotime was computed for a trajectory tree of aligned and integrated datasets in CAPITAL and the other tools, respectively. We will show the superiority of one method over its competitors from two viewpoints: biological variance

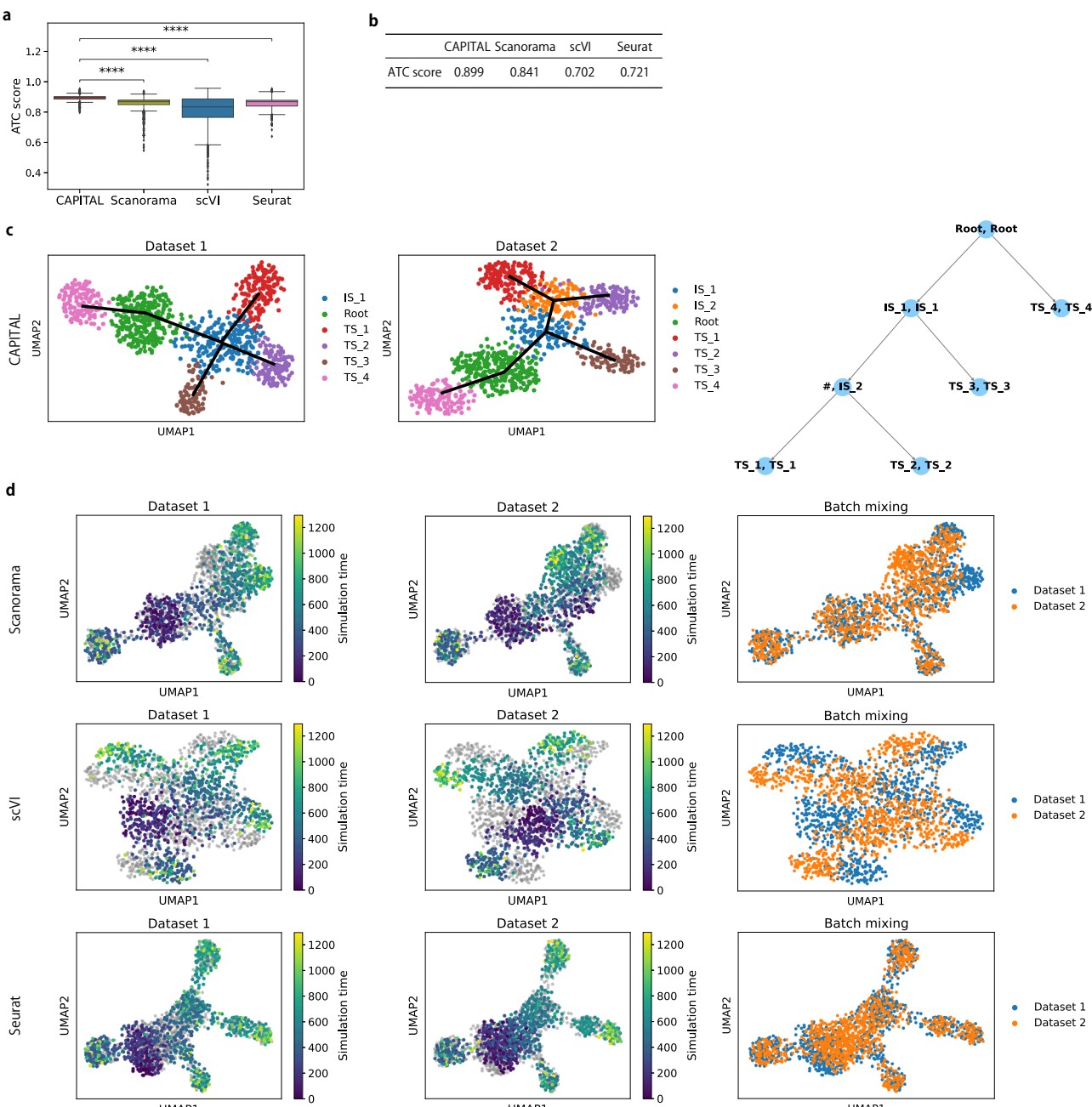

**Fig. 4 | Comparison of alignment accuracy between CAPITAL and data integration methods on a pair of synthetic datasets. a** Performance evaluation in trajectory conservation at the single-cell level on 2278 pairs of the synthetic datasets. A one-sided Wilcoxon singed-rank test was performed for each pair of the methods. ****indicates a *p*-value $< 1.00 \times 10^{-226}$ ($3.18 \times 10^{-272}$, $7.32 \times 10^{-239}$, and $7.76 \times 10^{-227}$ for CAPTAL vs. Scanorama, CAPITAL vs. scVI, and CAPITAL vs. Seurat, respectively), meaning that the ATC score of CAPITAL was significantly higher than those of the other tools. The box plots show the median with bounds for the first and the third quartiles, and the whiskers indicate the minimum and the maximum that exclude outliers represented by points. **b** ATC scores of all tools on datasets 1 and 2. **c** UMAP plots of datasets 1 and 2 with Leiden clustering, whose cell types

were annotated by considering simulation time and expression patterns of transcription factors (Supplementary Fig. 2). The solid lines indicate the trajectories. The rightmost column shows an aligned trajectory tree of those datasets predicted by CAPITAL. **d** UMAP plots of integration of datasets 1 and 2 computed by three data integration methods. The first and second columns indicate true simulation times in datasets 1 and 2, respectively, on the merged dataset, and the rightmost column shows UMAP plots of batch mixing. 10 nearest neighbors were considered to build the nearest neighbor graphs in all the tests. Source data are provided in the Source Data file. ATC average trajectory conservation, UMAP uniform manifold approximation and projection, IS intermediate state, TS terminal state.

conservation and batch removal before and after alignment/integration[16]. First, the results measured by the ATC score as a metric of biological variance conservation indicate that CAPITAL was statistically significantly better than the data integration approaches (Fig. 4a). In particular, CAPITAL was more robust to the variation of the datasets that contained multiple branches than the other algorithms. Second, we demonstrate two examples of datasets on which CAPITAL achieved

the most successful alignment, whereas the other algorithms failed to some degree or another (Fig. 4b–d and Supplementary Figs. 2, 3). Specifically, CAPITAL was able to match all initial and terminal states, while Scanorama and scVI were unsuccessful in aligning some initial and terminal states, and Seurat partly failed to match initial states. In the end, CAPITAL achieved major advances over current integration methods in trajectory conservation for complex trajectory trees.

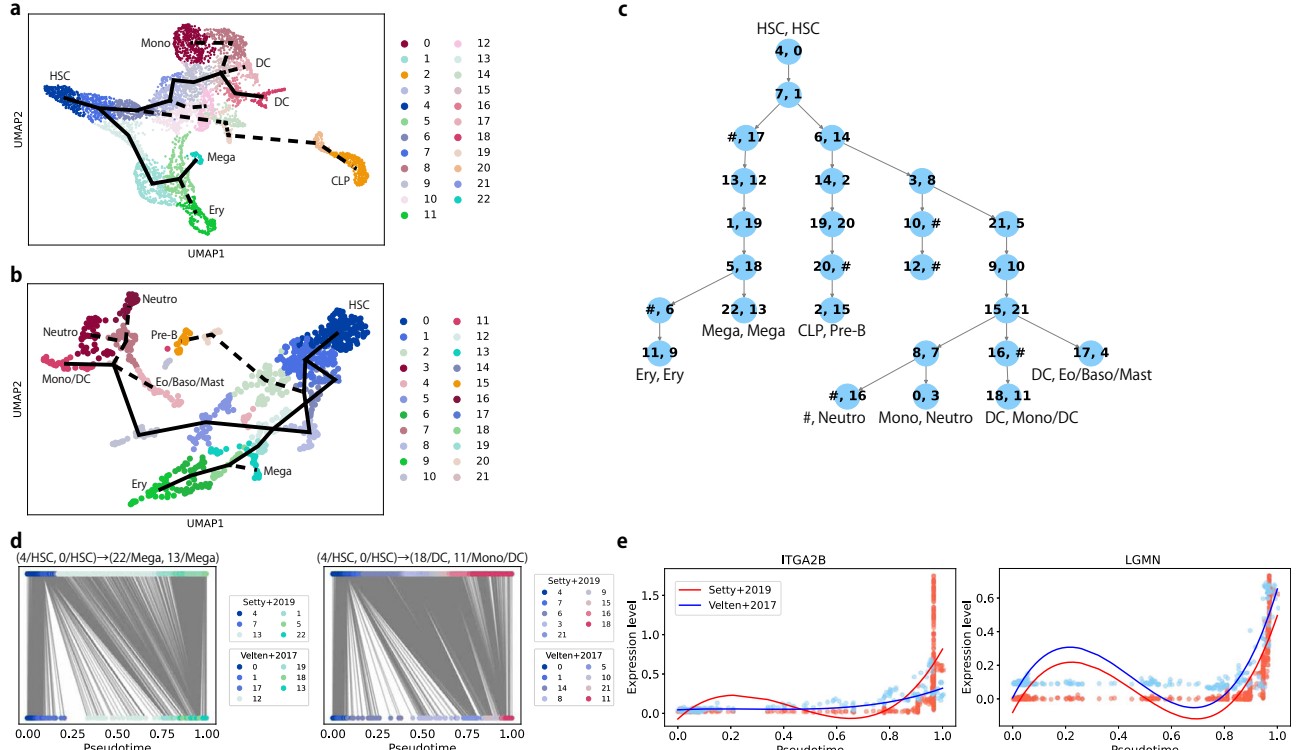

**Fig. 5 | Validation of CAPITAL on different scRNA-seq datasets of human bone marrow cells. a** A UMAP plot of clustering results of Setty et al.'s data. A trajectory tree predicted by CAPITAL is shown by a thick black line, where the solid lines indicate the linear trajectories that are compared in (**d**) and (**e**), whereas the dashed lines represent the others. **b** A UMAP plot of Velten et al.'s data. **c** An aligned tree of the trajectories of clusters, where each pair of numbers in a node denotes the clusters shown in (**a**) and (**b**). # denotes a space, meaning that it has no aligned cluster pair. **d** Cell−cell matchings along the trajectory paths from root (4/HSC, 0/ HSC) to (22/Mega, 13/Mega) and to (18/DC, 11/Mono/DC), respectively, which result from the alignment of the single cells in pseudotime order via dynamic time warping. Note that the cells in each dataset have the same color as the corresponding UMAP plot. **e** Pseudotime aligned kinetics for megakaryocyte and monocyte/dendritic cell markers. UMAP uniform manifold approximation and projection, HSC hematopoietic stem cell, CLP common lymphoid progenitor, DC dendritic cell, Eo/Baso eosinophil/basophil, Ery erythrocyte, Mega megakaryocyte, Mono monocyte, Neutro neutrophil.

## Alignment of differentiation trajectories in human bone marrow cells

To test how CAPITAL can compare two pseudotime trajectories with multiple branches generated from authentic scRNA-seq datasets, we used two public datasets of human bone marrow cells[17,18]. We ran CAPITAL to perform clustering of these scRNA-seq datasets on the basis of community detection in their nearest neighbor graphs, and illustrated the clustering results along with their trajectories in the two-dimensional space via uniform manifold approximation and projection (UMAP) (Fig. 5a, b and "Methods"). The two trajectory trees were then aligned, indicating that the algorithm was able to compute as many matching clusters as possible at leaves between different experiments while keeping their pseudotime structures (Fig. 5c). For validity of the matching clusters in the aligned trajectory tree with marker gene expression, see Supplementary Figs. 4–7. For instance, erythrocytes, megakaryocytes, pre-B-cells and dendritic cells were matched with each other between the two datasets.

Taking several paths that start from root (4/HSC, 0/HSC) and end with respective leaves with cell-type annotations from the aligned trajectory tree (Fig. 5c), we reconstructed pseudotime orderings of single cells contained in those paths by calculating diffusion pseudotime[2], and computed the matchings between cells along the orderings with dynamic time warping (Fig. 5d). Of note, dynamic time warping was performed with the intersection of highly variable genes in both datasets. On the basis of the results of dynamic time warping, we investigated pseudotime kinetics for marker genes (Fig. 5e and Supplementary Fig. 8), which indicates a similar tendency with respect to expression dynamics. These results tell us that CAPITAL was able to detect correct linear trajectories to be compared without manually

selecting them, suggesting that global comparison was successfully performed.

## Alignment of differentiation trajectories between human and mouse

We finally tested CAPITAL on scRNA-seq datasets across species. Specifically, we used again Velten et al.'s data of human bone marrow cells, while using scRNA-seq data of mouse bone marrow cells provided in an early study[19]. We selected this pair of the datasets because the number of cells compared with the counterpart was more balanced than the pair with Setty et al.'s human data (Supplementary Fig. 9). Figure 6a shows the results of clustering with a trajectory computed by CAPITAL for the mouse scRNA-seq data. Aligning the trajectory of the human data with that of the mouse data yielded an aligned tree shown in Fig. 6b, which were validated in part with marker genes in Supplementary Figs. 6, 7, 10 and 11. The matching clusters across species were found on the paths toward erythrocyte, monocyte and neutrophil, although the overall number of matching clusters was fewer than within the same species (Fig. 5c).

On the basis of the linear trajectory alignments that were obtained from the aligned tree (Fig. 6b), we investigated various molecular patterns in those three cell types along pseudotime via dynamic time warping ("Methods"). The results show that subclasses of marker genes for erythrocyte, monocyte, and neutrophil were likely to be conserved (Fig. 6c and Supplementary Fig. 12). In contrast, different molecular patterns were also observed, among which known marker genes of those cell types were included (Fig. 6d and Supplementary Fig. 13). More specifically, IRF7 and CSF1, which are known markers for monocyte and erythrocyte, respectively, were more highly expressed in human than in

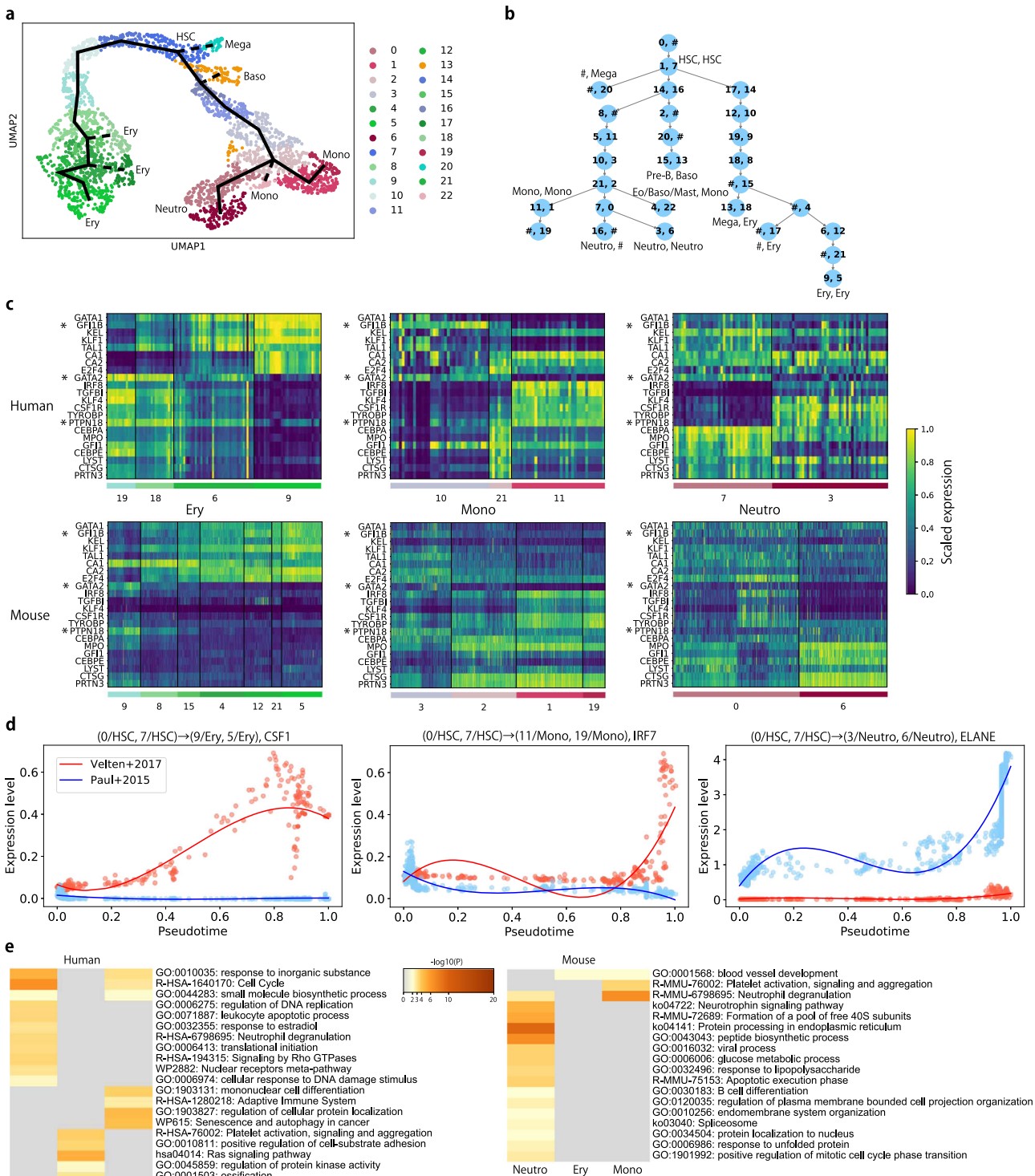

**Fig. 6 | Application of CAPITAL to cross-species scRNA-seq datasets of bone marrow cells. a** A UMAP plot of clustering results along with a trajectory inferred by CAPITAL for Paul et al.'s data. An estimated trajectory is shown by a thick black line, where the solid lines indicate the linear trajectories that are compared in (**c**–**e**), whereas the dashed lines represent the others. **b** An alignment of the trajectory trees, where each pair of numbers in a node denotes the clusters shown in Fig. 5b and this figure (**a**). # denotes a space. **c** Heatmaps of known cell-type markers for erythrocyte, monocyte, and neutrophil with similar expression patterns in the datasets. These are ordered by pseudotime from left to right in each heatmap. Note that the markers in mouse are orthologous to the ones in human, and for clarity, these names in mouse are exactly the same as those in the human dataset. A gene

name with an asterisk shows a marker that was not overlapped with the results of the computational screen ("Methods"). **d** Examples of different molecular patterns along pseudotime between human and mouse. **e** Heatmaps of enriched ontology terms across genes with different kinetics between human and mouse. They are colored by p-values computed by Metascape[20], where the one-sided statistical tests based on the hypergeometric distribution were performed. The left panel indicates the terms derived from genes that showed an increasing tendency of expression in the human dataset and a decreasing tendency in the mouse dataset, and the right panel vice versa. UMAP uniform manifold approximation and projection, HSC hematopoietic stem cell, Eo/Baso eosinophil/basophil, Ery erythrocyte, Mega megakaryocyte, Mono monocyte, Neutro neutrophil.

mouse as pseudotime went on, whereas ELANE, a marker for neutrophil, was more highly expressed in mouse than in human.

We further performed gene set enrichment analysis using Metascape[20]. For a set of genes that showed an increasing tendency of expression both in the human and the mouse datasets, a certain number of ontology terms specific to the respective mature cell types were found (Supplementary Fig. 14). This indicates that the genes computed by CAPITAL's framework were a valid set to investigate their kinetics. In comparison, the number of genes that showed the opposite kinetics between human and mouse was smaller than the similar kinetics, but a difference of ontology terms between human and mouse was observed to a large degree (Fig. 6e). Note that the difference shown here will result from technical issues such as cell cycle effects. Although more accumulated datasets across species are required to highlight differences in evolution, the use of CAPITAL would be one of the choices to detect a meaningful signal.

## Discussion

CAPITAL is a computational tool for comparing cell-state transition trajectory trees of different but related scRNA-seq datasets, aligning them globally without prior knowledge of linear paths to be selected. We implemented CAPITAL in Python, which can also be used in the interactive development environment JupyterLab, and evaluated its performance with exhaustive tests as compared with three data integration approaches. We also addressed the tasks of investigating various molecular patterns between different experiments on the same/different species through aligned pseudotime kinetics.

The problem of which clusters in two trajectory trees are being aligned is similar to the problem of selecting single lineages of single cells to be compared between the two trajectories, but they differ in computational complexity. The former first focuses on how to match clusters in the aligned tree followed by aligning single lineages of single cells that are uniquely determined, while the latter deals with single cells from scratch. If one would like to obtain the same/similar result using the above two approaches when no prior knowledge of the selection of single lineages is available, the latter requires all-against-all comparison of the single lineages in the trajectories. In contrast, CAPITAL can solve it de novo in a global fashion and run faster than the all-against-all alignment of the single lineages[8,9] (Supplementary Note 1).

Existing integration methods for matching cell clusters require the computation of anchors that link across different datasets to be compared, but such anchor-based data transformation still has a few limitations[21]. For example, there can be the wrong matching of cell subpopulations such as an integrated cluster with the imbalanced number of cells produced by data integration methods as demonstrated in our test. On the other hand, for the same pair of the datasets, CAPITAL was able to compute as many matching clusters as possible. This will be attributed to the difference of dimensionality when aligning datasets. For example, Scanorama and Seurat calculate mutual nearest neighbors in a low-dimensional space, whereas CAPITAL computes Spearman's correlation in high-dimensional expression space. Namely, the loss of information that is necessary for successful alignment might occur in the anchor-based integration in some cases.

In the tests with the three scRNA-seq datasets of bone marrow cells, we identified more than 20 clusters in each dataset, which were more than in the early studies[17–19]. These differences can be attributed to different preprocessing; in fact, we adopted a different way of processing data in our own framework. We confirmed that denoising and imputing expression counts of single cells were likely to generate more number of predicted clusters than previously reported. Most of the newly increased clusters could be considered as "intermediate states," which are intriguing per se since they are associated with the discovery of potential cell subtypes. Our results would suggest that CAPITAL showed the power to align trajectories composed of clusters beyond classical classification.

The results of cross-species analysis presented in this study might provide a promising methodology for identifying regulators of human disease. For instance, single-cell transcriptomes of normal and disease mouse models are experimentally sequenced, and their scRNA-seq datasets are compared to investigate aligned kinetics of candidate markers so that regulators for that disease can be revealed. These regulators in mouse are then computationally compared with the human scRNA-seq datasets of interest available in public databases, among which a subclass of regulators conserved between human and mouse might be found. Another application of cross-species analysis of scRNA-seq datasets is to elucidate the cellular diversity among multiple species. Similarly to multiple sequence alignment, which can be computed by the combination of pairwise alignments, sets of aligned trajectories of more than two species calculated by CAPITAL could be combined into one tree alignment that may include a common trajectory for multiple species.

We acknowledge that there are a few limitations in our study. First, part of the CAPITAL algorithm for computing a trajectory as a minimum spanning tree may not be the best method. However, we remind that the core part of CAPITAL is to align cell-state transition trajectory trees. In this sense, trajectory trees predicted by any method can readily be incorporated into CAPITAL, meaning that it can employ a state-of-the-art or even a future tool with higher predictive performance. Second, alignment accuracy of CAPITAL for datasets with different global topological structure degraded in our benchmarking test. To resolve this issue, designing an algorithm based on local tree alignment or tree inclusion[22] might be a promising approach. Third, CAPITAL in its present form can deal only with trajectory trees (including single lineages) that are computed from input datasets. Indeed, complex trajectories beyond tree such as cycle and disconnected graph will appear in transdifferentiation. Hence, a predictive model that can deal with a graph beyond tree[5,23] along with graph alignment[24] will be required to further develop techniques for complex comparative pseudotime analysis.

CAPITAL's framework can be applied to not only pseudotemporal data but also spatial and epigenetic data. Given that scRNA-seq techniques have prevailed in current research in cell biology, and increasing attention has been paid to the comparative analysis of cellular processes with different conditions or species, we envision that CAPITAL will contribute to the advancement of comparative single-cell omics.

## Methods

### CAPITAL algorithm

**Clustering cells via graph structure.** Let $X = (x_1, \ldots, x_n)^\top \in \mathbb{R}^{n \times m}$ be an expression count matrix of $m$ genes across $n$ cells, where $x_j = (x_{1j}, \ldots, x_{mj})^\top \in \mathbb{R}^m (1 \le j \le n)$ is the logarithm of a normalized count vector of cell $j$ with respect to genes $1, \ldots, m$. Note that log transform is performed with one pseudocount, and the genes are often assumed to be highly variable in expression over all cells. To reduce the dimensionality of the data, principal component analysis (PCA) is performed on $X$, resulting in $Z = (z_1, \ldots, z_n)^\top \in \mathbb{R}^{n \times p}$, where $p(\ll m)$ is the number of principal components that contribute to the total variance in the data. A weighted graph is constructed in a way that each vertex represents a cell, and each edge connecting two cells $i$ and $j$ has a weight calculated by the Euclidian distance $\|z_i - z_j\|$ between the two cells. From this graph, a $k$-nearest neighbor ($k$-NN) graph can be built on the basis of the distance assigned to each edge, as it can better capture phenotypic relatedness[25]. Clustering vertices in the $k$-NN graph is then performed by a community detection method such as the Leiden algorithm[14].

**Inferring a pseudotime trajectory.** To make the $k$-NN graph structure simple, a cluster *centroid* is defined as the virtual cell with the medians of $p$ principal components derived from those of the cells belonging to

that cluster. A minimum spanning tree in this centroid graph is then computed with Kruskal's algorithm. If one chooses one of the centroids as the root of the tree, the resulting rooted minimum spanning tree can be regarded as a pseudotime trajectory of the clusters. Of note, actual pseudotime of each single cell along a path in the tree will be estimated later.

**Preliminaries to handling trees.** Let $T = (V, E)$ be an unordered labeled rooted tree, where $V$ and $E$ (also denoted by $V(T)$ and $E(T)$) are a set of nodes (vertices) and that of edges of the tree, respectively, and the children of each node are regarded as a set. The number of nodes in tree $T$ is represented as $|T|$. Let $\theta$ be the empty tree, and let $T(i)$ be the subtree of $T$ induced by node $i$ and all of its descendants. If node $i$ has children $i_1, \dots, i_\mu$, i.e. the degree of node $i$ is $\mu$, we define $F(i) = F(i_1, \dots, i_\mu)$ as the forest comprising subtrees $T(i_1), \dots, T(i_\mu)$.

Inserting node $w$ as a child of node $v \in V(T)$ makes $w$ be the parent of a consecutive subset of the children of node $v$. An alignment of unordered labeled trees $T_1$ and $T_2$ is obtained by inserting nodes labeled with *spaces#* into $T_1$ and/or $T_2$ so that the two trees are isomorphic if the labels are ignored, and then by overlaying the resulting trees. This means that # is regarded as the empty node. Let $\gamma : (\Sigma_\#, \Sigma_\#) \setminus (\{\#\}, \{\#\}) \to \mathbb{R}$ denote a metric cost function on pairs of labels, where $\Sigma$ is a finite alphabet and $\Sigma_\# = \Sigma \cup \{\#\}$. We often extend this notation to nodes so that $\gamma(i, j)$ means $\gamma(\text{label}(i), \text{label}(j))$ for $i, j \in V(T)$. For scRNA-seq data analysis, we define the cost function as

$$\gamma(i,j) = \begin{cases} 1 - \text{corr}(\boldsymbol{x}_i, \boldsymbol{x}_j) & (\text{label}(i) \text{ and label}(j) \text{ correspond to cells } i \text{ and } j, \text{ respectively}), \\ \delta & (\text{label}(i) = \# \text{ or label}(j) = \#), \end{cases} \tag{1}$$

where $\text{corr}(\boldsymbol{x}_i, \boldsymbol{x}_j)$ returns Spearman's rank correlation coefficient between expression levels $\boldsymbol{x}_i$ and $\boldsymbol{x}_j$ of cells $i$ and $j$, respectively, and $\delta$ is some constant. In all computational tests in this study, we set $\delta = 1$.

The cost of the alignment is defined as the sum of the costs of all paired labels in the alignment. An *optimal alignment* of $T_1$ and $T_2$ is an alignment with the minimum cost, which is called the optimal *alignment distance* between $T_1$ and $T_2$, denoted by $D(T_1, T_2)$. This notion can be extended to two forests $F_1$ and $F_2$, denoted by $D(F_1, F_2)$.

**Tree alignment algorithm.** The problem of computing an optimal alignment distance between general unordered trees is MAX SNP-hard[26]. However, aligning unordered trees with "bounded" degrees can be solved in polynomial time. In what follows, we will summarize the dynamic programming (DP) algorithm for calculating the optimal alignment distance between unordered trees with bounded degrees presented in the literature[26].

The initial settings for handling the empty tree during the DP calculation are defined as follows:

$$D(\theta, \theta) = 0, \tag{2}$$

$$D(T_1(i), \theta) = D(F_1(i), \theta) + \gamma(i, \#), \tag{3}$$

$$D(F_1(i), \theta) = \sum_{k=1}^{\mu} D(T_1(i_k), \theta), \tag{4}$$

$$D(\theta, T_2(j)) = D(\theta, F_2(j)) + \gamma(\#, j), \tag{5}$$

$$D(\theta, F_2(j)) = \sum_{k=1}^{\nu} D(\theta, T_2(j_k)), \tag{6}$$

where $\mu$ and $\nu$ are degrees of nodes $i \in V(T_1)$ and $j \in V(T_2)$, respectively.

We next look at the computation of an optimal alignment distance between the trees.

$$D(T_1(i), T_2(j)) = \min \begin{cases} D(F_1(i), F_2(j)) + \gamma(i, j), \\ D(T_1(i), \theta) + \min_{1 \le r \le \mu} \{ D(T_1(i_r), T_2(j)) - D(T_1(i_r), \theta) \}, \\ D(\theta, T_2(j)) + \min_{1 \le r \le \nu} \{ D(T_1(i), T_2(j_r)) - D(\theta, T_2(j_r)) \}. \end{cases} \tag{7}$$

This recursion means that there are three cases to be considered for the DP calculation: (i) paired nodes $(i, j)$ is in the alignment; (ii) $(i, \#)$ and $(i_r, j)$ for some child of node $i$ are in the alignment; and (iii) $(\#, j)$ and $(i, j_r)$ for some child of node $j$ are in the alignment (Supplementary Fig. 15).

Finally, we focus on how to compute an optimal alignment distance $D(F_1(i), F_2(j))$ between the forests that appeared in Equation (7). Since all combinations of forest pairs derived from unordered trees $T_1(i)$ and $T_2(j)$ need to be considered, we define subsets of the forests denoted by $\mathcal{A} \subseteq \{T_1(i_1), \dots, T_1(i_\mu)\}$ and $\mathcal{B} \subseteq \{T_2(j_1), \dots, T_2(j_\nu)\}$. The alignment distance between the forest sets $\mathcal{A}$ and $\mathcal{B}$ can then be computed by

$$D(\mathcal{A}, \mathcal{B}) = \min \begin{cases} \min_{T_1(i_p) \in \mathcal{A}, T_2(j_q) \in \mathcal{B}} \{ D(\mathcal{A} - T_1(i_p), \mathcal{B} - T_2(j_q)) + D(T_1(i_p), T_2(j_q)) \}, \\ \min_{T_1(i_p) \in \mathcal{A}, \mathcal{B}' \subseteq \mathcal{B}} \{ D(F_1(i_p), \mathcal{B}') + D(\mathcal{A} - T_1(i_p), \mathcal{B} - \mathcal{B}') + \gamma(i_p, \#) \}, \\ \min_{\mathcal{A}' \subseteq \mathcal{A}, T_2(j_q) \in \mathcal{B}} \{ D(\mathcal{A}', F_2(j_q)) + D(\mathcal{A} - \mathcal{A}', \mathcal{B} - T_2(j_q)) + \gamma(\#, j_q) \}, \end{cases} \tag{8}$$

where $1 \le p \le \mu$ and $1 \le q \le \nu$ (Supplementary Fig. 16). Note that $D(\mathcal{A}, \mathcal{B})$ such that $\mathcal{A} = \{T_1(i_1), \dots, T_1(i_\mu)\}$ and $\mathcal{B} = \{T_2(j_1), \dots, T_2(j_\nu)\}$ is equivalent to $D(F_1(i), F_2(j))$. Since degrees $\mu$ and $\nu$ are bounded and regarded as constants in this case, time and space complexities of the tree alignment algorithm are evaluated as $O(|T_1||T_2|)$. The pseudocode for running the tree alignment algorithm is described in Supplementary Note 2.

**Pseudotime ordering.** Having been obtained from a trajectory alignment between two datasets, a pair of aligned paths is arbitrarily chosen, and all single cells in the clusters are to be processed in each dataset. Note that the starting cell in the first cluster on the aligned path has to be determined for the subsequent pseudotime ordering in a way that it has the longest distance to a cell among all other cells in the corresponding dataset in the expression space. A diffusion pseudotime in each dataset is then calculated by taking into consideration an accumulated transition matrix for all cells with respect to the path based on a random walk on the nearest neighbor graph[2].

**Dynamic time warping.** Dynamic time warping is an algorithm for measuring similarity between temporal sequences. Briefly, given two time series, the algorithm aims to find the best matching between the two sequences by stretching or compressing elements of the sequences. For optimization, the distance between two elements across the sequences measured on the basis of time warping functions is summed over all elements, which is then to be minimized by dynamic programming. Details of the algorithm can be found in the literature[10].

## A metric for matching clusters

Let $M$ and $U$ be the number of matched cluster pairs and that of unmatched cluster pairs, respectively, in an aligned trajectory tree. The optimal alignment distance $D(T_1, T_2)$ for trajectory trees $T_1$ and $T_2$ can be decomposed into the sum of the distances for the matched cluster pairs and that for the unmatched cluster pairs, i.e. $D(T_1, T_2) = d_M + d_U$.

The normalized alignment distance is then defined by

$$d_{\text{norm}} = \alpha \frac{d_M}{M+1} + (1-\alpha) \frac{d_U}{U+1}, \qquad (9)$$

where $\alpha (0 < \alpha < 1)$ is a balancing parameter between the contribution of the matched clusters and that of the unmatched ones. Note that weight $\alpha$ for the per-node distance of the matched cluster pairs should be higher than $1-\alpha$ for that of the unmatched cluster pairs because $0 < d_M/(M+1) \le d_U/(U+1) < d_U/U = U/U = 1$ holds asymptotically when cost $\gamma$ is determined by comparing matching cost $1 - \text{corr}()$ with gap penalty $\delta = 1$ as defined above. Namely, the latter penalty term in the normalized alignment distance needs to be smaller than the former essential term of the alignment distance for the matched clusters. We set $\alpha = 0.9$ in the tests in feasibility and robustness of CAPITAL (Figs. 2 and 3a). A short normalized alignment distance is regarded as an accurate alignment in cluster matching.

## A metric for aligning trajectory trees

We define a parameter-free metric on an aligned trajectory tree of two scRNA-seq datasets to evaluate biological variation conservation before and after alignment, which is called an average trajectory conservation (ATC) score. Trajectory conservation was originally defined in the literature[16], and in this work we adapt the definition to our case where the tree structure should be explicitly considered to compute pseudotime. Precisely, for each single lineage that forms the aligned trajectory tree, a series of simulation times of the cells generated by dyngen[15] along that lineage is regarded as a reference, while a series of pseudotimes of alined/integrated datasets as a prediction, between which Spearman's rank correlation coefficient is calculated. Note that a starting cell of alined/integrated datasets for all tools, which is necessary to compute diffusion pseudotime, is defined in a way that its simulation time in dyngen is 0 and it has the longest distance to a cell among all other cells in the expression space. In the case of CAPITAL, single-cell alignment is performed by dynamic time warping with two series of pseudotimes (e.g. for datasets 1 and 2), and thus the mean of the two respective correlation coefficients between the series of simulation times and either series of pseudotimes should be taken. Each Spearman's rank correlation coefficient $\rho$ computed above is scaled to the range [0, 1] by $(\rho + 1)/2$, which we call a trajectory conservation score of a single lineage. Finally, the ATC score of the aligned trajectory tree is defined as the mean of the trajectory conservation scores of all single lineages. A high ATC score is considered to be an accurate alignment in trajectory conservation.

## Creating synthetic datasets

First, a gene regulatory network with the backbone as a binary tree with three branches for 1000 cells was generated by dyngen 1.0.3[15]. In this step, 36 transcription factors were generated, and 250 target genes and 250 housekeeping genes were sampled. Second, simulating kinetics, a gold standard, and cells 120 times resulted in 120 different datasets of single-cell expression count matrices. Finally, these 120 datasets were filtered in such a way that the number of leaves in a minimum spanning tree based on the clustering results is exactly four, as expected in the simulation, which leads to 68 synthetic datasets of expression count matrices. For benchmarking CAPITAL's performance, each dataset was randomly split into two disjoint count matrices 100 times, resulting in $68 \times 100 = 6800$ pairs of count matrices to be tested.

## Adding noise

Gaussian noise was independently added 100 times to each of the 6800 pairs of the log transformed count matrices obtained above, where the parameters of the distribution were set to zero mean and increasing standard deviation from 0 to 5.0 by step size 0.5, followed

by adding a larger step size to reach 50.0 standard deviation. Note that the alignments of the 6800 pairs without noise (i.e. zero mean and zero standard deviation) can be regarded as the true alignments, and thus comparison between the true alignment and the predicted alignment on the data with noise makes sense. To summarize, $6800 \times 14 = 95,200$ pairs were used in the test for robustness (Fig. 3).

## Comparison with data integration methods

To compare the performance of other methods for integrating scRNA-seq datasets, we used Scanorama 1.7.0[12] and scVI 0.16.0[11] as wrapper functions of single-cell integration benchmark (scib 1.0.3)[16], and Seurat 4.1.1[13] to infer a common trajectory of each of all 2278 pairs of the 68 synthetic datasets described above. Of note, we used synthetic datasets for the comparison, as real datasets do not have shared ground truth to validate trajectory alignments. All pairs of the datasets in the form of either raw counts or preprocessed counts were integrated by the respective algorithms. A common trajectory was then estimated by finding a minimum spanning tree whose nodes were centroids as defined in the CAPITAL algorithm, which was considered as an aligned trajectory tree. Note that the same methods and the parameters as in CAPITAL were used to compute neighborhood graphs, Leiden clustering, and diffusion pseudotime.

## Preprocessing data

Before aligning different scRNA-seq datasets, each may need to be preprocessed due to noise such as outlier cells included in the original data. All real datasets used in this work were filtered in advance by keeping cells with at least 200 genes expressed and genes that were expressed in at least three cells, which were performed with Scanpy 1.9.1[27]. In what follows, we will describe how to adapt all datasets to the subsequent trajectory comparison.

**Synthetic datasets.** In each synthetic dataset, the top 200 highly variable genes were used to obtain 50 principal components, from which a 10-NN graph was built for cell clustering and trajectory inference. Note that these settings were common to the tests on all tools.

**Setty et al.'s data.** A single-cell expression count dataset of human bone marrow cells was downloaded at https://github.com/dpeerlab/Palantir. More precisely, the Scanpy AnnData object[27] of replicate 1 with 5780 cells, which was generated in the existing study[17], was used to preprocess in our downstream analysis. MAGIC[28] for denoising and imputation was applied to the logarithmic normalized counts. From the resulting count matrix, 50 principal components derived from the top 2000 highly variable genes were used to construct a 40-NN graph, leading to clusters and a trajectory tree (Fig. 5a and Supplementary Fig. 5b). In particular, clusters were annotated by investigating the expression of marker genes for each cell type (Supplementary Figs. 4 and 5a). The root of the tree was set to cluster 4 (HSC) as marker CD34 was highly expressed in the cluster.

**Velten et al.'s data.** A normalized filtered single-cell count dataset of 1034 human bone marrow cells (individual 1) was downloaded from Gene Expression Omnibus (GSE75478 [https://www.ncbi.nlm.nih.gov/geo/query/acc.cgi?acc=GSE75478]). Note that these data were normalized with posterior odds ratio[18] and included negative values due to its definition. After the common filtering of outlier cells and genes described above, cells were further removed as they still had too many genes or too many total counts. Specifically, cells with $m$ genes and $c$ total normalized counts were retained such that $4600 < m < 5750$ and $-2000 < c < 1500$, leading to 915 cells. MAGIC was applied to the resulting count matrix, min-max scaling was performed so that each element in the matrix was non-negative, and the elements were log transformed with one pseudocount. The top 250 highly variable genes were used to derive 50 principal components, which was used to build

a 6-NN graph. Community detection and trajectory inference on this graph yielded the results shown in Fig. 5b and Supplementary Fig. 7b. With cluster annotations using marker genes for each cell type (Supplementary Figs. 6 and 7a), cluster 0 (HSC) was set to the root of the trajectory tree as marker HOXA3 was highly expressed.

**Paul et al.'s data.** A single-cell dataset of mouse bone marrow cells[19] was obtained from one of the datasets accessible when running Scanpy[27]. This dataset was first preprocessed according to Scanpy's tutorial with Zheng et al.'s preprocessing recipe[29]. It is to be noted that this preprocessing contained dimensionality reduction with diffusion map. A coarse-grained PAGA graph[5] derived from the Louvain clustering of the preprocessed data had two disconnected outlier clusters, which were removed to have 2671 cells in our demonstration so that CAPITAL can deal with them. To perform further denoising, MAGIC was used on the above count matrix with each element normalized and log transformed. 50 principal components computed from the top 500 highly variable genes were used to construct a 10-NN graph, on which clustering with annotation and trajectory inference were carried out (Fig. 6a and Supplementary Fig. 11b). Some of the clusters were annotated as specific cell types as was done in the human bone marrow cells described above (Supplementary Figs. 10 and 11a). Cluster 7 (HSC) was estimated as one of the candidate roots of the trajectory tree by taking into account the expression of Meis1 and Itga2b, which was also discussed in the early work[19].

### Analysis of molecular patterns

First, in the aligned tree for Velten et al.'s data and Paul et al.'s data (Fig. 6b), highly variable genes in either dataset for each aligned path were selected in a way that they had a larger normalized dispersion than a predefined cutoff[27], which was set to 1.0 in this study. It should be noted that those genes were independently selected per branch in the aligned tree. Second, similarity scores for the kinetics of the respective genes in the linear alignments were computed via dynamic time warping. The genes were then split into three groups by considering their inclination of linear regression lines so that both kinetics curves sloped up, one curve sloped up but the other down, and vice versa. Finally, the genes in each group were filtered by set difference to identify potential cell type-specific genes (Supplementary Figs. 12 and 13). Note that the likely conserved gene sets shown in Fig. 6c were selected among known markers to validate the results as compared with prior knowledge, but we confirmed that most of them overlapped with the genes that were computed by the above method.

### Statistics and reproducibility

No statistical method was used to predetermine sample size. Some outlier cells in isolated clusters in the original public data were excluded according to pre-established criteria. The experiments were not randomized. The Investigators were not blinded to allocation during experiments and outcome assessment.

### Reporting summary

Further information on research design is available in the Nature Research Reporting Summary linked to this article.

### Data availability

The original single-cell expression count dataset of human bone marrow cells provided by Setty et al. is available at https://github.com/dpeerlab/Palantir. The other human bone marrow cells dataset compiled by Velten et al. is available from Gene Expression Omnibus (GSE75478 [https://www.ncbi.nlm.nih.gov/geo/query/acc.cgi?acc=GSE75478]). The mouse dataset provided by Paul et al. is accessible when running the datasets.paul15() function in Scanpy (1.9.1). The synthetic datasets, and human and mouse cell datasets preprocessed in this study are accessible through a GitHub repository[30] [https://

github.com/ykat0/capital]. They are also available in a Code Ocean compute capsule [https://codeocean.com/capsule/5673663/tree/v1]. All other relevant data supporting the key findings of this study are available within the article and its Supplementary Information files or from the corresponding author upon reasonable request. Source data are provided with this paper.

### Code availability

CAPITAL is implemented with Python, making good use of a single-cell analysis toolkit Scanpy[27]. In particular, CAPITAL can be used in an interactive development environment such as JupyterLab. CAPITAL codes and documentation with tutorials are available through the GitHub repository[30] [https://github.com/ykat0/capital] as well as the Code Ocean compute capsule [https://codeocean.com/capsule/5673663/tree/v1].

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

## Acknowledgements

This work was supported by the Japan Society for the Promotion of Science KAKENHI [21K12109 to Y.Kato; 19K20399 to T.Mori]. This study was partly achieved through the use of large-scale computer systems at the Cybermedia Center, Osaka University, and the NIG supercomputer at ROIS National Institute of Genetics.

## Author contributions

Y.Kato and T.M. conceived the work. R.S., Y.Kato, and T.M. designed and discussed the algorithms. R.S and Y.Kato wrote the CAPITAL codes, and performed all computational experiments. R.S., Y.Kato, and Y.Kawahara discussed the results. All authors contributed to the scope of the work, revised and approved the final manuscript.

## Competing interests

The authors declare no competing interests.
