## [Peer Review File · Nature Communications]

Editorial Note: This manuscript has been previously reviewed at another journal that is not operating a transparent peer review scheme. This document only contains reviewer comments and rebuttal letters for versions considered at Nature Communications

REVIEWER COMMENTS

Reviewer #1 (Remarks to the Author):

Summary

The authors present a novel method for aligning trajectories between datasets with multiple branching points. Their approach relies on matching minimum spanning trees between clustered datasets using Spearman correlations and a dynamic programming algorithm. Matching trajectories across datasets is an interesting and impactful problem in single-cell analysis that can aid in the comparison of cellular differentiation across species and datasets. Current approaches to trajectory alignment do not directly deal with branching trajectories, but require manual selection of linear trajectory segments. This is solved by the presented algorithm.

Overall, the manuscript includes a detailed description of the method, an extensive evaluation on simulated data, and two case study applications to real scRNA-seq datasets. While the application of a tree matching algorithm to this problem enables a solution without a manual selection of trajectories, the biological impact of this novelty is not clear. It seems that any interpretation of aligned trajectories would require the same manual input for selecting trajectories that this algorithm automates. Furthermore, the comparison with alternative approaches (data integration) does not currently highlight a clear use case for the presented approach. Highlighting such use cases would vastly improve the manuscript and increase its impact.

Major comments

Pseudotime is not an absolute scale that can be compared between trajectories. Yet the authors propose to compare pseudotime between species. It is unclear to me whether "regulation timing for an orthologous gene" can be achieved without an absolute baseline scale.

How do you assess that the clustering in both datasets is similar or was done properly? In particular, clustering resolution should affect the tree structure and therefore the ability of CAPITAL to align the datasets.

No mention is made regarding global topological similarity in the assumptions. It would be good to assess whether CAPITAL works when parts of the data are removed that affect the global topological structure. Outliers were removed in preprocessing for the Paul et al data, but it is not clear why. Could the authors provide a breaking example here to highlight when the algorithm works and when it doesn't? This would be invaluable for users.

What is a "sufficient level" for noise robustness? And how does decreasing scores with increasing noise suggest that CAPITAL is robust to noise addition? How does the added noise compare to noise that is expected in a typical dataset?

The comparison between data integration and CAPITAL is not entirely convincing. The authors argue that data integration and trajectory inference may generate a wrong

matching of subpopulations. While this can happen in data integration, why should it not also occur when matching medoids on the basis of spearman correlation if no exact matching is available? Some additional analyses would strengthen this section:

Seurat and CAPITAL performance looks incredibly similar in the metric computed. How frequently is Seurat worse/better/within a 0.05 ABWAP score margin in the 1,373 simulation comparisons?

Is there a particular example you can highlight where data integration does not perform well, but CAPITAL does? Is there an example of the opposite?

How does the ABWAP score change with differing penalty terms for (c, #)?

It would be good to have some validation of the cluster alignment in Figure 2c. Could the authors show marker gene expression across these clusters in a dotplot to highlight that aligned clusters are indeed correctly matched? This is especially crucial as the authors indicate their higher number of clusters is down to an improved pre-processing. This statement would need to be validated by annotation of the clusters with meaningful labels.

Please highlight the sections of the tree in Fig 2a,b that are being compared in Figure 2d,e. It is currently not clear if all branches are included in the comparison or not. Some may explain the Setty et al., data's pseudotime arm around 0.8 in Figure 2e.

Building a minimum spanning tree only on the basis of cluster medoids (called centroid in the paper) leads to effects such as trees that don't match the transitions hinted at by the single-cell distributions. For example in Figure 2b cluster 5 seems to connect clusters 2 and 6 (if I'm reading the colours correctly), but the tree links it to the myeloid lineage. It would be good if this plot was annotated via the original labels so that readers can assess whether the inferred tree is true to the underlying differentiation processes.

What is a "relative temporal lag peculiar to each marker"? Pseudotime values do not have to be comparable between different datasets as they do not have to linearly map to an inherent time scale, but can be distorted.

Do you achieve similar results for cross-species trajectory mapping for the Velten and Setty et al data? It would be good to have some assessment of method robustness to build confidence in the method.

How were the conserved gene sets across species selected? Could you include other important markers such as GATA2 for the erythroid lineage, MKI67 to highlight intermediate proerythroblasts, and MECOM for monocyte/neutrophil progenitors.

The manuscript states that the different terms that are enriched may highlight differences in evolution. I wonder what terms the authors are referring to. The enrichment results appear to highlight technical differences such as cell cycle effects, and apoptosis. Differences of genes that relate to neutrophil degranulation may indicate that one dataset included a stronger inflammatory response in the cells. Some discussion of these differences would be useful to assess whether a meaningful signal was found in the cross-species comparison.

The authors mention that leiden clustering gives between 5 and 10 clusters in the simulations. However, the resolution for this clustering is not mentioned. How does the algorithm respond to different clustering resolutions?

How many cells did the authors filter out from the Velten et al data? The range of allowed numbers of genes seems very narrow. Typically one would keep cells with ~500-1000 genes and not filter out cells with high numbers of genes. Is there any reason for this?

How were highly variable genes selected per cell type if no cluster annotation was performed? Furthermore, a HVG of a neutrophil would not necessarily capture the differentiation process of neutrophils. Or were the genes selected HVG in all clusters of the neutrophil lineage?

While I could find the notebooks that reproduce several of the paper figures, for which I commend the authors, I could not find any notebook detailing how the dataset preprocessing was done, how the heatmap genes were selected and this figure was generated, or showing the comparison between Seurat and CAPITAL. It would be great if these were added.

Biological interpretation of the aligned trajectory trees requires the reader to understand which clusters are being aligned in the trajectory to interpret the gene expression dynamics that are being compared. This seems analogous to selecting a linear segment of the trajectory tree to compare with former approaches to trajectory alignment. Thus, the practical innovation of CAPITAL over, for example, cellAlign is not clear to me. Could the authors clarify this? Maybe a comparison with cellAlign or the Cacchiarelli et al. trajectory alignment approach would be helpful?

Minor comments

typo: "patters" -> "patterns"

remove embellishing words like "actually"

typo "dynamic time warming" -> "dynamic time warping"

Figure 2d caption: what is a cell-cell matching for a cell marker? Please clarify the caption

It would be easiest for the reader to separate the 3 lineages in Fig 3c into separate plots (erythroid, monocyte, and neutrophil). These would ideally also be ordered by pseudotime to assess whether the inferred dynamics match the expression patterns.

"were highly expressed in human than..." -> "were more highly expressed in human than...". Same with "was highly expressed in mouse than" -> "was more highly..." below

"between the forests appeared" -> "between the forests that appeared"

"are needed to be considered" -> "need to be considered"

which packages/platforms were used for data preprocessing? Seurat? If so, please cite and mention the version.

Please add an explanation of what "denoising with diffusion map" is. This is not clear to me.

Reviewer #2 (Remarks to the Author):

1. it is still not clear how much improvement CAPITAL provides over the naïve approach of merging the two datasets together and creating a shared branched trajectory. The present approach might be similar or even inferior to the integration of cells between the datasets and the inference of the branches afterwards

2. In Supp. Figure 1, I would expect CAPITAL would reach ABWAP score of around zero at

very high noise levels. However, it seems like there is not so much of a decrease at all (maybe a very slow one). This tells me that the score might not be efficient to test the method's accuracy. I suspect that this also affects the comparison between CAPITAL and Seurat where the first obtained only a marginal improvement at best. One possible solution would be creating one single cell data and then dividing it into different pairs. Different levels of noise can be added to each of the divided data. In that case, the true alignment is known so a measure of distance between the true and the predicted alignment can be assessed.

3. In the gene enrichment analysis, it is not clear how and why genes are selected per cell type if the alignment is done per branch. Please elaborate on that.

Responses to Reviewer #1's comments

Thank you very much for reading our manuscript carefully and giving the helpful comments. We list our responses in roman type to the reviewer's comments italicized. Please note that we have added some sentences, figures and references in the main text because the early manuscript that were transferred from *Nature Biotechnology* followed the instruction for "Brief Communication," which imposed more strict word, figure and reference limitations than "Articles" in *Nature Communications*. For example, we have added some of the figures for new computational tests in this revision and of the previous supplementary figures in the main text to strengthen the results obtained in this study. We have also added two paragraphs to Discussion section to explain the advantage and the limitations of this work so that readers can better understand our study.

Summary

The authors present a novel method for aligning trajectories between datasets with multiple branching points. Their approach relies on matching minimum spanning trees between clustered datasets using spearman correlations and a dynamic programming algorithm. Matching trajectories across datasets is an interesting and impactful problem in single-cell analysis that can aid in the comparison of cellular differentiation across species and datasets. Current approaches to trajectory alignment do not directly deal with branching trajectories, but require manual selection of linear trajectory segments. This is solved by the presented algorithm.

Overall, the manuscript includes a detailed description of the method, an extensive evaluation on simulated data, and two case study applications to real scRNA-seq datasets. While the application of a tree matching algorithm to this problem enables a solution without a manual selection of trajectories, the biological impact of this novelty is not clear. It seems that any interpretation of aligned trajectories would require the same manual input for selecting trajectories that this algorithm automates. Furthermore, the comparison with alternative approaches (data integration) does not currently highlight a clear use case for the presented approach. Highlighting such use cases would vastly improve the manuscript and increase its impact.

Response. Thank you very much for your interest in our work. We have addressed all the points raised by the two reviewers. To the best of our knowledge, no methods for aligning single-cell trajectories can deal with branching topology so far. If one uses an existing linear alignment method to align branching trajectories, all-against-all comparison of the linear segments is required when no prior information for selecting paths is available. In contrast, our approach can solve it de novo in a global fashion and run faster than the naive all-against-all linear alignment. We think that enabling de novo global comparison of single-cell trajectories will have a biological impact since an unexpected discovery without prior knowledge can be made in the future, namely will be

advantageous to align completely unannotated datasets (please see also our response to comment (17)). Moreover, by responding to major comments (2)–(5) and (13), which required additional computational tests, we believe that our paper has been much improved to convince the novelty and the advantage of this work.

Major

- (1) *Pseudotime is not an absolute scale that can be compared between trajectories. Yet the authors propose to compare pseudotime between species. It is unclear to me whether “regulation timing for an orthologous gene” can be achieved without an absolute baseline scale.*

Response. Indeed pseudotime is not an absolute baseline scale, but we think it possible to compare between single-cell trajectories of different datasets by determining one trajectory as a reference and the other as a query in the dynamic time warping algorithm [1]. In this approach, pseudotimes in both datasets are rescaled to the range from 0 to 1, between which each cell in one dataset is aligned to a cell in the other. This indirectly makes the comparison possible as performed in early studies [2, 3]. If one would like to quantify a relative temporal lag of a gene of a specific time point, pseudotime shift between expression dynamics can be calculated, which was proposed in the literature [2]. Please note that the pseudotime shift does not have biological meaning but makes sense when making comparison. In this work, we did not focused on a specific time point and thus we just compared the whole trend of pseudotime between datasets as shown in Fig. 5d.

- (2) *How do you assess that the clustering in both datasets is similar or was done properly? In particular, clustering resolution should affect the tree structure and therefore the ability of CAPITAL to align the datasets.*

Response. First, we have defined the normalized alignment distance to assess the similarity of clustering between two datasets as follows (Methods section on page 19):

Let M and U be the number of matched cluster pairs and that of unmatched cluster pairs, respectively, in an aligned trajectory tree. The optimal alignment distance $D(T_1, T_2)$ for trajectory trees T_1 and T_2 can be decomposed into the sum of the distances for the matched cluster pairs and that for the unmatched cluster pairs, i.e. $D(T_1, T_2) = d_M + d_U$. The normalized alignment distance is then defined by

$$d_{\text{norm}} = \alpha \frac{d_M}{M + 1} + (1 - \alpha) \frac{d_U}{U + 1},$$

where α ($0 < \alpha < 1$) is a balancing parameter between the contribution of the matched clusters and that of the unmatched ones. Note that

weight α for the per-node distance of the matched cluster pairs should be higher than $1 - \alpha$ for that of the unmatched cluster pairs because $0 < d_M/(M + 1) \leq d_U/(U + 1) < d_U/U = U/U = 1$ holds asymptotically when cost γ is determined by comparing matching cost $1 - \text{corr}$ with gap penalty $\delta = 1$. Namely, the latter penalty term in the normalized alignment distance needs to be smaller than the former essential term of the alignment distance for the matched clusters. We set $\alpha = 0.9$ in the tests in feasibility and robustness of CAPITAL, while using different values of α to check the effect on the normalized alignment distance.

We have then performed a test in clustering similarity on 31,800 pairs of trajectories generated from our synthetic datasets according to the reviewer’s suggestion, whose results have been described in Results section on page 5 (Fig. 2b):

Next, the clustering dissimilarity between two datasets tended to scale linearly with the normalized alignment distance, indicating that the clustering similarity affected the performance of the subsequent trajectory alignment (Fig. R1).

Fig. R1. A test in clustering similarity on 31,800 pairs of trajectories. 15 nearest neighbors were considered to build the nearest neighbor graphs. Weight $\alpha = 0.9$ for the matched clusters to calculate the normalized alignment distance was used. Each box shows the quartiles of the distribution, and the whiskers indicate the rest of the distribution except for the points that represent outliers.

- (3) *No mention is made regarding global topological similarity in the assumptions. It would be good to assess whether CAPITAL works when parts of the data are removed that affect the global topological structure. Outliers were removed in preprocessing for the Paul et al data, but it is not clear why. Could the authors provide a breaking example here to highlight when the algorithm works and when it doesn't? This would be invaluable for users.*

Response. We apologize for the missing assumption of global topological similarity. We have described the effect of global topological similarity on the performance of CAPITAL in Results section on pages 4–5 (Fig. 2a):

In the first test to show the effect of global topological similarity between two datasets, the algorithm worked from the viewpoint of the normalized alignment distance (Methods) even when minor changes were made to one of the trajectory trees to be aligned (e.g. removing a leaf, internal node, etc) (Fig. R2). In contrast, a breaking example of removing the root with two children, which destroys a global topological similarity between two trajectories, shows that the normalized alignment distance clearly increased (the alignment performance declined).

Fig. R2. A test in global topological similarity on 38,193 pairs of trajectories. Reference shows the results of trajectory alignments with no nodes in one trajectory tree removed. The other cases where a node was removed in one trajectory tree are classified into five types. 15 nearest neighbors were considered to build the nearest neighbor graphs. Weight $\alpha = 0.9$ for the matched clusters to calculate the normalized alignment distance was used. Each box shows the quartiles of the distribution, and the whiskers indicate the rest of the distribution except for the points that represent outliers.

Please note that we removed outlier cells in Paul *et al.*'s data because they were disconnected cells in the coarse-grained graph that resulted from following Scanpy's tutorial [4], and CAPITAL can deal with connected acyclic graph (i.e. tree) (see also Methods section on pages 22–23).

- (4) *What is a "sufficient level" for noise robustness? And how does decreasing scores with increasing noise suggest that CAPITAL is robust to noise addition? How does the added noise compare to noise that is expected in a typical dataset?*

Response. We apologize for any ambiguity regarding the description in the previous manuscript. According to the other reviewer’s suggestion, we have performed a robustness test with newly generated datasets. The way of creating the new noise-added datasets is explained in Methods section on pages 20–21:

Gaussian noise was independently added 100 times to each of the 5,300 pairs of the log transformed count matrices, where the parameters of the distribution were set to zero mean and increasing standard deviation from 0 to 5.0 by step size 0.5. Note that the alignments of the 5,300 pairs without noise (i.e. zero mean and zero standard deviation) can be regarded as the true alignments, and thus comparison between the true alignment and the predicted alignment on the data with noise makes sense. To summarize, $5,300 \times 11 = 58,300$ pairs were used in the test for robustness.

Fig. R3. A performance indicator of CAPITAL as a function of the noise level on the synthetic datasets. The horizontal axis indicates the standard deviation of Gaussian noise that was independently added 100 times to each of the expression counts obtained by splitting a single-cell count matrix in the 53 synthetic datasets, which results in 58,300 pairs of trajectories. **a**, Performance evaluation in cluster–cluster alignment. Weight $\alpha = 0.9$ for the matched clusters to calculate the normalized alignment distance was used. **b**, Performance evaluation in cell–cell alignment. Penalty $\beta = 0.6$ for each unmatched cluster to compute the ABWAP score was used. 15 nearest neighbors were considered to build the nearest neighbor graphs in all the tests. Each box shows the quartiles of the distribution, and the whiskers indicate the rest of the distribution except for the points that represent outliers.

Using these datasets, we have tested the robustness of CAPITAL with both the normalized alignment distance and the ABWAP score in Results section on pages 5–6 (Fig. 3):

Second, we tested the robustness of CAPITAL measured by alignment accuracy on the synthetic datasets with increasing data noise. More pre-

cisely, we evaluated the alignment accuracy from two measures: (i) the normalized alignment distance for assessing the performance of cluster–cluster alignment; and (ii) a metric defined on aligned trees based on area between worst and prediction (ABWAP) at a single-cell level. The rate of change in the normalized alignment distance was higher for the noise level of at least 3.5 than at most 3.0 (Fig. R3a), and that of the ABWAP score was higher for the noise level of at least 3.0 than at most 2.5 (Fig. R3b). Given that the noise of standard deviation 5.0 is unlikely to emanate from a typical dataset (e.g. standard deviation 5.0 was much larger than average 0.64 of the non-zero elements in the noise-free count matrices in our simulation), these results suggest that CAPITAL was robust to data noise to a certain degree at both a cluster-matching level and a single-cell alignment level.

For additional information on how big the noise of standard deviation 5.0 is, please see the following figure that shows PCA plots of the Leiden clustering of the same dataset to which Gaussian noise of standard deviation $\sigma \in \{1.0, 3.0, 5.0\}$ was added. These results indicate that clustering almost worked when adding noise of standard deviation 3.0, while shattered for noise of standard deviation 5.0.

Fig. R4. PCA plots of clustering results of a synthetic dataset. The number in parentheses that follows each graph title shows the standard deviation of Gaussian noise added to the data.

- (5) *The comparison between data integration and CAPITAL is not entirely convincing. The authors argue that data integration and trajectory inference may generate a wrong matching of subpopulations. While this can happen in data integration, why should it not also occur when matching medoids on the basis of spearman correlation if no exact matching is available? Some additional analyses would strengthen this section: Seurat and CAPITAL performance looks incredibly similar in the metric computed. How frequently is Seurat worse/better/within a 0.05 ABWAP score margin in the 1,373 simulation comparisons? Is there a particular example you can highlight where data integration does not perform well, but CAPITAL does? Is there an example of the opposite? How does the ABWAP score change with differing penalty terms for $(c, \#)$?*

Response. We agree that the previous comparison between CAPITAL and Seurat was not convincing. To make the difference in their alignment performance apparent, we have reanalyzed the results according to the reviewer’s suggestion.

First, we have added results using the normalized alignment distance as well as changing the parameters appeared in the definitions of the alignment distance and the ABWAP score. These are shown in Results section on pages 6–7 (Fig. 4a,b), and Supplementary Tables 1 and 2:

Finally, to compare CAPITAL with a method of data integration in alignment performance, we ran Seurat [5] on the combinations of the synthetic datasets to merge two respective datasets with anchors and perform common trajectory inference on that merged dataset. Note that data integration across datasets was not performed in CAPITAL. We show that the superiority of one method over its competitor with respect to the normalized alignment distance depended on the value of balancing parameter α assigned to the matched clusters against the unmatched ones in an aligned tree (Fig. R5a and Table R1). This means that CAPITAL outperformed Seurat when putting emphasis on the matched clusters, while Seurat had an advantage when the unmatched clusters increased in number. Next, the results measured by the ABWAP score indicate that CAPITAL was statistically better than Seurat when penalty β for each unmatched cluster exceeded 0.5 (Fig. R5b and Table R2). Given that the ABWAP score takes into account each of the cell orders to be aligned rather than just a cluster-matching quality measured by the alignment distance, one may say that CAPITAL could be better suited than Seurat to perform a subclass of the tasks of aligning single-cell trajectories.

Table R1. Frequencies with respect to the normalized alignment distance for comparisons between CAPITAL and Seurat on 1,378 pairs of the synthetic datasets. “CAPITAL < Seurat” means that CAPITAL has normalized alignment distances 0.05 smaller than Seurat does, and “CAPITAL > Seurat” for the opposite case. “CAPITAL \approx Seurat” shows that they have normalized alignment distances within a 0.05 margin as compared with each other.

Weight	CAPITAL < Seurat	CAPITAL > Seurat	CAPITAL \approx Seurat
0.80	314	474	590
0.85	319	378	681
0.90	347	286	745
0.95	426	235	717

Table R2. Frequencies with respect to the ABWAP score for comparisons between CAPITAL and Seurat on 1,371 pairs of the synthetic datasets. “CAPITAL > Seurat” means that CAPITAL has ABWAP scores 0.05 higher than Seurat does, and “CAPITAL < Seurat” for the opposite case. “CAPITAL \approx Seurat” shows that they have ABWAP scores within a 0.05 margin as compared with each other.

Penalty	CAPITAL > Seurat	CAPITAL < Seurat	CAPITAL \approx Seurat
0.4	386	432	555
0.5	374	356	643
0.6	352	273	748
0.7	321	201	851
0.8	323	174	876

Fig. R5. Comparison of alignment accuracy between CAPITAL and Seurat on the synthetic datasets. **a**, Performance evaluation in cluster–cluster alignment on 1,378 pairs of the synthetic datasets. A number shown in parentheses on the horizontal axis is weight α assigned to the matched clusters to compute the normalized alignment distance. A one-sided Wilcoxon signed-rank test was performed for each pair of the methods with the same weight. * indicates that CAPITAL was statistically smaller in distance than Seurat, whereas † shows vice versa. ††: a p-value $< 1.00 \times 10^{-7}$; †: a p-value < 0.05 ; *: a p-value < 0.01 ; **: a p-value $< 1.00 \times 10^{-14}$. **b**, Performance evaluation in cell–cell alignment on 1,371 pairs of the synthetic datasets. A number shown in parentheses on the horizontal axis is penalty β assigned to each unmatched cluster in an aligned tree to calculate the ABWAP score. A one-sided Wilcoxon signed-rank test was performed for each pair of the methods with the same penalty. * indicates that the ABWAP score of CAPITAL was statistically higher than that of Seurat. n.s.: not significant; *: a p-value < 0.01 ; **: a p-value $< 1.00 \times 10^{-6}$; ***: a p-value $< 1.00 \times 10^{-12}$. 15 nearest neighbors were considered to build the nearest neighbor graphs in all the tests. Each box shows the quartiles of the distribution, and the whiskers indicate the rest of the distribution except for the points that represent outliers. **c**, Bar plots of the ratio of cells that belong to aligned clusters between two example datasets where CAPITAL computed better than Seurat did ($\alpha = 0.9$ and $\beta = 0.6$). **d**, An opposite case of c with two different datasets ($\alpha = 0.9$ and $\beta = 0.6$).

Second, we have highlighted two examples where CAPITAL computed better than Seurat did and vice versa in Results section on pages 6–7 (Fig. 4c,d) along with Supplementary Figs. 1 and 2:

We further looked more carefully into a difference in alignment results between CAPITAL and Seurat. As one example, CAPITAL computed an alignment of two datasets better than Seurat did. Specifically, Seurat did not properly integrate the datasets, so that its alignment result was worse than that of CAPITAL (Fig. R5c). In contrast, Seurat calculated a good alignment of two other datasets that were successfully integrated, although CAPITAL yielded a passable result (Fig. R5d). These examples would suggest an advantage of CAPITAL over Seurat when aligning two trajectories in some cases.

Finally, one of the reasons why the data integration approach failed in some cases has been discussed in Discussion section on page 12:

Existing integration methods for matching cell clusters require the computation of anchors that link across different datasets to be compared, but such anchor-based data transformation still has a few limitations [6]. For example, there can be the wrong matching of cell subpopulations such as an integrated cluster with the imbalanced number of cells produced by Seurat as demonstrated in our test. On the other hand, for the same pair of the datasets, CAPITAL was able to compute as many matching clusters as possible. This will be attributed to the difference of dimensionality when aligning datasets: Seurat calculates mutual nearest neighbors in a low-dimensional space, whereas CAPITAL computes Spearman's correlation in a high-dimensional expression space. Namely, the loss of information that is necessary to successful alignment might occur in the anchor-based integration in some cases.

Please note that “overcorrection” mentioned in the previous manuscript could be made in the non-linear batch correction algorithm [6], but this was not true to our case demonstrated in this paper. Thus, we have rewritten the paragraph as shown above.

- (6) *It would be good to have some validation of the cluster alignment in Figure 2c. Could the authors show marker gene expression across these clusters in a dotplot to highlight that aligned clusters are indeed correctly matched? This is especially crucial as the authors indicate their higher number of clusters is down to an improved pre-processing. This statement would need to be validated by annotation of the clusters with meaningful labels.*

Response. We have added the dot plots that show marker gene expression across clusters for Setty *et al.*'s data, Velten *et al.*'s data and Paul *et al.*'s data as Supplementary Figs. 4, 6 and 9, respectively. Considering the results, we think that aligned clusters were correctly matched to a large degree, and have added the phrase to the corresponding part on pages 8–9.

Fig. R6. A dot plot with respect to marker genes and a trajectory tree of Setty *et al.*'s data.
a, A dot plot of scaled expression levels of marker genes across clusters for Setty *et al.*'s data. The number appearing on the left side in the plot corresponds to the node number shown in **b**.
b, A trajectory tree of Setty *et al.*'s data predicted by CAPITAL. The number in a node in the tree is equivalent to the cluster number of the same color shown in Fig. 5a in the main text.

Fig. R7. A dot plot with respect to marker genes and a trajectory tree of Velten *et al.*'s data. a, A dot plot of scaled expression levels of marker genes across clusters for Velten *et al.*'s data. The number appearing on the left side in the plot corresponds to the node number shown in b. **b,** A trajectory tree of Velten *et al.*'s data predicted by CAPITAL. The number in a node in the tree is equivalent to the cluster number of the same color shown in Fig. 5b in the main text.

Fig. R8. A dot plot with respect to marker genes and a trajectory tree of Paul *et al.*'s data.
a, A dot plot of scaled expression levels of marker genes across clusters for Paul *et al.*'s data. The number appearing on the left side in the plot corresponds to the node number shown in **b**.
b, A trajectory tree of Paul *et al.*'s data predicted by CAPITAL. The number in a node in the tree is equivalent to the cluster number of the same color shown in Fig. 6a in the main text.

- (7) *Please highlight the sections of the tree in Fig 2a,b that are being compared in Figure 2d,e. It is currently not clear if all branches are included in the comparison or not. Some may explain the Setty et al. data's pseudotime arm around 0.8 in Figure 2e.*

Response. First, we have changed the representation in Fig. 5a,b on page 8 in a way that solid lines indicate linear trajectories that were compared in Fig. 5d,e, whereas dashed lines represent the others. This change has also been made to Fig. 6a on page 10.

Second, we have added an explanation for the pseudotime arm that the reviewer points out in Results section on page 9:

Note that the pseudotime arm around 0.8 in Setty *et al.*'s data did not result from imputation as the majority of the plots with the same condition in Supplementary Fig. 7 did not have outliers.

- (8) *Building a minimum spanning tree only on the basis of cluster medoids (called centroid in the paper) leads to effects such as trees that don't match the transitions hinted at by the single-cell distributions. For example in Figure 2b cluster 5 seems to connect clusters 2 and 6 (if I'm reading the colours correctly), but the tree links it to the myeloid lineage. It would be good if this plot was annotated via the original labels so that readers can assess whether the inferred tree is true to the underlying differentiation processes.*

Response. We agree that a cluster-based trajectory tree does not always reflect the transitions of single cells. Please note that cluster 5 in the center of Fig. 5b directly connects clusters 8 and 10 of the similar color as illustrated in Supplementary Fig. 6b, and we do not think it as problematic.

We also agree that annotations of the plots using original labels would be of interest. However, we feel that these analyses make us identify a cell type in the original clustering using a dot plot of marker genes, which is essentially equivalent to what we have done using the dot plots shown in Supplementary Figs. 4, 6 and 9.

- (9) *What is a "relative temporal lag peculiar to each marker"? Pseudotime values do not have to be comparable between different datasets as they do not have to linearly map to an inherent time scale, but can be distorted.*

Response. We have added the following sentence to the original main text on page 9 to answer the question (please see also our response to comment (1)):

As shown in the figure, cell–cell matching does not necessarily agree with the corresponding cluster–cluster alignment due to the relative temporal lag peculiar to each marker, which can be found by dynamic time warping with scaled pseudotime.

(10) Do you achieve similar results for cross-species trajectory mapping for the Velten and Setty *et al* data? It would be good to have some assessment of method robustness to build confidence in the method.

Fig. R9. An alignment of the trajectory trees, where each pair of numbers in a node denotes the clusters shown in Fig. 5a and Fig. 6a. # denotes a space. HSC, hematopoietic stem cell; CLP, common lymphoid progenitor; DC, dendritic cell; Ery, erythrocyte; Mega, megakaryocyte; Mono, monocyte; Neutro, neutrophil.

Response. We guess that the reviewer refers to the comparison between Setty *et al.*'s human data and Paul *et al.*'s mouse data. We have tried aligning this pair with CAPITAL, but unfortunately it was unsuccessful in terms of the matching clusters with the same annotation (Fig. R9). Specifically, the stem cell clusters (7, 7) and the erythrocyte clusters (11, 4) were matched with each other, but the other cell types were not matched. We should not overlook that the difference (3,109) in the number of cells in Setty(5,780)–Paul(2,671) datasets is about 1.8 times as large as that (1,756) in Velten(915)–Paul(2,671) datasets, where each number in parentheses that follows the author name shows the number of cells in that dataset. Given that both the numbers of clusters in the two datasets are the same (Figs. 5a and 6a), each cluster centroid in Paul *et al.*'s data would have less information of gene expression than in Setty *et al.*'s data, which might result in the unsuccessful alignment. Thus, we have not shown this result in the main text, but added the following sentence on page 9:

We selected this pair of the datasets because the number of cells compared with the counterpart was more balanced than the pair with Setty

et al.'s human data.

- (11) *How were the conserved gene sets across species selected? Could you include other important markers such as GATA2 for the erythroid lineage, MKI67 to highlight intermediate proerythroblasts, and MECOM for monocyte/neutrophil progenitors?*

Response. We selected the likely conserved gene sets shown in Fig. 6c among known markers to validate the results as compared with prior knowledge, but we also confirmed that most of them overlapped with the genes that were computed by the computational screen (please see Methods section on page 23). In this revision, we have added GATA2, and separated the heatmap into the three cell-type paths as suggested in minor comment (5). Please note that GATA2, MKI67 and MECOM were not in the results from the unsupervised computational screen mentioned above for Velten *et al.*'s human data, and MKI67 and MECOM were not included in Paul *et al.*'s mouse data obtained from the dataset accessible when running Scanpy [4].

- (12) *The manuscript states that the different terms that are enriched may highlight differences in evolution. I wonder what terms the authors are referring to. The enrichment results appear to highlight technical differences such as cell cycle effects, and apoptosis. Differences of genes that relate to neutrophil degranulation may indicate that one dataset included a stronger inflammatory response in the cells. Some discussion of these differences would be useful to assess whether a meaningful signal was found in the cross-species comparison.*

Response. We apologize for exaggerating the results of this work toward differences in evolution. As the reviewer pointed out, we acknowledge that the enrichment results were likely to highlight technical differences, and share the reviewer's curiosity on this point. Unfortunately, the existing datasets to reveal a cross-species difference are limited. Hence, we have added the following sentences in Results section on page 11:

Note that the difference shown here will result from technical issues such as cell cycle effects. Although more accumulated datasets across species are required to highlight differences in evolution, use of CAPITAL would be one of the choices to detect a meaningful signal.

- (13) *The authors mention that leiden clustering gives between 5 and 10 clusters in the simulations. However, the resolution for this clustering is not mentioned. How does the algorithm respond to different clustering resolutions?*

Response. We apologize for the missing information on resolution. We used the Leiden clustering of resolution one to create our synthetic datasets, and have added this in Methods section on page 20.

Next, we have investigated the alignment performance of CAPITAL when changing the number of nearest neighbors in Results section on page 5. Please note that changing the number of nearest neighbors used yields a similar result obtained by changing the clustering resolution.

The third test in clustering resolution tells us that the number of nearest neighbors used to construct a trajectory has to be carefully chosen (Fig. R10). Note that considering the larger number of nearest neighbors was likely to yield the smaller number of clusters (i.e. lower resolution), namely the alignment of trajectories with the larger number of clusters was more difficult than the opposite case. Taken together, suitable clustering for building a trajectory per dataset including choice of the number of nearest neighbors is necessary to enhance alignment performance of CAPITAL.

Fig. R10. A test in clustering resolution on 31,800 pairs of trajectories. Weight $\alpha = 0.9$ for the matched clusters to calculate the normalized alignment distance was used. Each box shows the quartiles of the distribution, and the whiskers indicate the rest of the distribution except for the points that represent outliers.

(14) *How many cells did the authors filter out from the Velten et al data? The range of allowed numbers of genes seems very narrow. Typically one would keep cells with 500-1000 genes and not filter out cells with high numbers of genes. Is there any reason for this?*

Response. We agree with the reviewer on the typical preprocessing method for scRNA-seq data. We observed, however, that the cells in Velten *et al.*'s data that were preprocessed in the standard method were so badly clustered that the subsequent analysis including inference of a trajectory tree was not successful. Besides, reproduction of the original clustering results presented in the paper was very hard

due to no clustering-annotated data available. Thus, we removed 119 poor-quality cells from 1,034 cells in the original data by the criterion stated in Methods section, which we do not think as a drastic reduction. In the revised manuscript, we have added the original number of cells in Methods section on page 22. The reason for the narrow number of genes kept is the same as above.

- (15) *How were highly variable genes selected per cell type if no cluster annotation was performed? Furthermore, a HVG of a neutrophil would not necessarily capture the differentiation process of neutrophils. Or were the genes selected HVG in all clusters of the neutrophil lineage?*

Response. We selected highly variable genes per branch (lineage) in the aligned trajectory tree. We have rephrased the corresponding statements in Methods section on page 23.

- (16) *While I could find the notebooks that reproduce several of the paper figures, for which I commend the authors, I could not find any notebook detailing how the dataset preprocessing was done, how the heatmap genes were selected and this figure was generated, or showing the comparison between Seurat and CAPITAL. It would be great if these were added.*

Response. We apologize for the incomplete disclosure in Code Ocean. We have added the notebooks for preprocessing the three authentic datasets, drawing the expression trends and heatmaps shown in the main text, and comparing CAPITAL with Seurat. Please note that clustering results may be different in each computational platform because the Leiden algorithm has a random factor and will affect the result even if a seed is fixed [7]. This is why we provided preprocessed data in Scanpy's AnnData format [4] to ensure the reproducibility. As for the testing performance of the tools with our synthetic datasets, it required huge computational resources such as those in super computers to align over 1,000 trajectories. Thus, we have only uploaded the results (e.g. CSV files) that were computed on a super computer system to draw some figures in the main text.

- (17) *Biological interpretation of the aligned trajectory trees requires the reader to understand which clusters are being aligned in the trajectory to interpret the gene expression dynamics that are being compared. This seems analogous to selecting a linear segment of the trajectory tree to compare with former approaches to trajectory alignment. Thus, the practical innovation of CAPITAL over, for example, cellAlign is not clear to me. Could the authors clarify this? Maybe a comparison with cellAlign or the Cacchiarelli et al. trajectory alignment approach would be helpful?*

Response. We agree that the problem of which clusters in two trajectory trees are being aligned is similar to the problem of selecting linear segments of single

cells to be compared between the two trajectories, but they differ in computational complexity. The details have been discussed in Discussion section on pages 11–12:

The problem of which clusters in two trajectory trees are being aligned is similar to the problem of selecting linear segments of single cells to be compared between the two trajectories, but they differ in computational complexity. The former first focuses on how to match clusters in the aligned tree followed by aligning linear segments of single cells that are uniquely determined, while the latter deals with single cells from scratch. If one would like to obtain the same/similar result using the above two approaches when no prior knowledge of the selection of linear segments is available, the latter requires all-against-all comparison of the linear segments in the trajectories. In contrast, CAPITAL can solve it de novo in a global fashion and run faster than the all-against-all alignment of the linear segments [2, 3] (Supplementary Note 1).

Thus, we do not think that computational tests in comparison between CAPITAL and a linear alignment method such as cellAlign [2] would make sense. Of course, CAPITAL can incorporate an alternative linear alignment method other than dynamic time warping [1] into the latter part of the algorithm (cell-based linear alignment), and we hope to clarify the performance in a future study.

Minor

(1) *typo: “patters” → “patterns”*

Response. We have corrected the typo.

(2) *Remove embellishing words like “actually.”*

Response. We have removed those embellishing words.

(3) *typo “dynamic time warming” → “dynamic time warping”*

Response. We have corrected the typo.

(4) *Figure 2d caption: what is a cell–cell matching for a cell marker? Please clarify the caption.*

Response. We have rephrased the corresponding sentences as follows:

Cell–cell matchings for erythrocyte and monocyte/dendritic cell markers along the trajectory paths from root (4/HSC, 0/HSC) to (11/Ery, 9/Ery) and to (18/DC, 11/Mono/DC), respectively, which result from alignment of the single cells in pseudotime order via dynamic time warping. Note

that some of the linear alignments with respect to the specific genes are shown for display, and the cells in each dataset have the same color as the corresponding UMAP plot.

- (5) *It would be easiest for the reader to separate the 3 lineages in Fig 3c into separate plots (erythroid, monocyte, and neutrophil). These would ideally also be ordered by pseudotime to assess whether the inferred dynamics match the expression patterns.*

Response. We have separated the heatmap into the three lineages for each organism. Please note that the new figures are slightly different from the previous ones due to the scaled expression that were computed from different length of paths in the aligned tree, but we have come to the same conclusion.

- (6) *“were highly expressed in human than...” → “were more highly expressed in human than...”. Same with “was highly expressed in mouse than” → “was more highly...” below.*

Response. We have rewritten these phrases as indicated.

- (7) *“between the forests appeared” → “between the forests that appeared”*

Response. We have corrected this phrase as indicated.

- (8) *“are needed to be considered” → “need to be considered”*

Response. We have rewritten this phrase as indicated.

- (9) *Which packages/platforms were used for data preprocessing? Seurat? If so, please cite and mention the version.*

Response. We used Scanpy ver. 1.9.1 [4] to preprocess all the datasets used in this work because it allows seamless integration with CAPITAL in the Python framework. We have added the description of the tool for preprocessing in Methods section on page 21.

- (10) *Please add an explanation of what “denoising with diffusion map” is. This is not clear to me.*

Response. We apologize for the ambiguous statement. We used the word “denoising” to mean dimensionality reduction, and thus we have rephrased it as “dimensionality reduction with diffusion map” on page 22.

Responses to Reviewer #2's comments

Thank you very much again for reading our revised manuscript and giving the helpful comments. We list our responses in roman type to the reviewer's comments italicized. Please note that we have added some sentences, figures and references in the main text because the early manuscript that were transferred from *Nature Biotechnology* followed the instruction for "Brief Communication," which imposed more strict word, figure and reference limitations than "Articles" in *Nature Communications*. For example, we have added some of the figures for new computational tests in this revision and of the previous supplementary figures in the main text to strengthen the results obtained in this study. We have also added two paragraphs to Discussion section to explain the advantage and the limitations of this work so that readers can better understand our study.

Major

- (1) *It is still not clear how much improvement CAPITAL provides over the naive approach of merging the two datasets together and creating a shared branched trajectory. The present approach might be similar or even inferior to the integration of cells between the datasets and the inference of the branches afterwards.*

Response. We agree that the previous comparison between CAPITAL and the naive approach with Seurat was not clear. To make the difference in their alignment performance apparent, we have reanalyzed the results according to the other reviewer's suggestion.

First, we have added results using the normalized alignment distance as well as changing the parameters appeared in the definitions of the alignment distance and the ABWAP score. The normalized alignment distance has been defined in Methods section on page 19:

Let M and U be the number of matched cluster pairs and that of unmatched cluster pairs, respectively, in an aligned trajectory tree. The optimal alignment distance $D(T_1, T_2)$ for trajectory trees T_1 and T_2 can be decomposed into the sum of the distances for the matched cluster pairs and that for the unmatched cluster pairs, i.e. $D(T_1, T_2) = d_M + d_U$. The normalized alignment distance is then defined by

$$d_{\text{norm}} = \alpha \frac{d_M}{M+1} + (1 - \alpha) \frac{d_U}{U+1},$$

where α ($0 < \alpha < 1$) is a balancing parameter between the contribution of the matched clusters and that of the unmatched ones. Note that weight α for the per-node distance of the matched cluster pairs should be higher than $1 - \alpha$ for that of the unmatched cluster pairs because

$0 < d_M/(M + 1) \leq d_U/(U + 1) < d_U/U = U/U = 1$ holds asymptotically when cost γ is determined by comparing matching cost $1 - \text{corr}$ with gap penalty $\delta = 1$. Namely, the latter penalty term in the normalized alignment distance needs to be smaller than the former essential term of the alignment distance for the matched clusters. We set $\alpha = 0.9$ in the tests in feasibility and robustness of CAPITAL, while using different values of α to check the effect on the normalized alignment distance.

The results are shown in Results section on pages 6–7 (Fig. 4a,b), and Supplementary Tables 1 and 2:

Finally, to compare CAPITAL with a method of data integration in alignment performance, we ran Seurat [5] on the combinations of the synthetic datasets to merge two respective datasets with anchors and perform common trajectory inference on that merged dataset. Note that data integration across datasets was not performed in CAPITAL. We show that the superiority of one method over its competitor with respect to the normalized alignment distance depended on the value of balancing parameter α assigned to the matched clusters against the unmatched ones in an aligned tree (Fig. R5a and Table R1). This means that CAPITAL outperformed Seurat when putting emphasis on the matched clusters, while Seurat had an advantage when the unmatched clusters increased in number. Next, the results measured by the ABWAP score indicate that CAPITAL was statistically better than Seurat when penalty β for each unmatched cluster exceeded 0.5 (Fig. R5b and Table R2). Given that the ABWAP score takes into account each of the cell orders to be aligned rather than just a cluster-matching quality measured by the alignment distance, one may say that CAPITAL could be better suited than Seurat to perform a subclass of the tasks of aligning single-cell trajectories.

Table R1. Frequencies with respect to the normalized alignment distance for comparisons between CAPITAL and Seurat on 1,378 pairs of the synthetic datasets. “CAPITAL < Seurat” means that CAPITAL has normalized alignment distances 0.05 smaller than Seurat does, and “CAPITAL > Seurat” for the opposite case. “CAPITAL \approx Seurat” shows that they have normalized alignment distances within a 0.05 margin as compared with each other.

Weight	CAPITAL < Seurat	CAPITAL > Seurat	CAPITAL \approx Seurat
0.80	314	474	590
0.85	319	378	681
0.90	347	286	745
0.95	426	235	717

Table R2. Frequencies with respect to the ABWAP score for comparisons between CAPITAL and Seurat on 1,371 pairs of the synthetic datasets. “CAPITAL > Seurat” means that CAPITAL has ABWAP scores 0.05 higher than Seurat does, and “CAPITAL < Seurat” for the opposite case. “CAPITAL \approx Seurat” shows that they have ABWAP scores within a 0.05 margin as compared with each other.

Penalty	CAPITAL > Seurat	CAPITAL < Seurat	CAPITAL \approx Seurat
0.4	386	432	555
0.5	374	356	643
0.6	352	273	748
0.7	321	201	851
0.8	323	174	876

Fig. R5. Comparison of alignment accuracy between CAPITAL and Seurat on the synthetic datasets. **a**, Performance evaluation in cluster–cluster alignment on 1,378 pairs of the synthetic datasets. A number shown in parentheses on the horizontal axis is weight α assigned to the matched clusters to compute the normalized alignment distance. A one-sided Wilcoxon signed-rank test was performed for each pair of the methods with the same weight. * indicates that CAPITAL was statistically smaller in distance than Seurat, whereas † shows vice versa. ††: a p-value $< 1.00 \times 10^{-7}$; †: a p-value < 0.05 ; *: a p-value < 0.01 ; **: a p-value $< 1.00 \times 10^{-14}$. **b**, Performance evaluation in cell–cell alignment on 1,371 pairs of the synthetic datasets. A number shown in parentheses on the horizontal axis is penalty β assigned to each unmatched cluster in an aligned tree to calculate the ABWAP score. A one-sided Wilcoxon signed-rank test was performed for each pair of the methods with the same penalty. * indicates that the ABWAP score of CAPITAL was statistically higher than that of Seurat. n.s.: not significant; *: a p-value < 0.01 ; **: a p-value $< 1.00 \times 10^{-6}$; ***: a p-value $< 1.00 \times 10^{-12}$. 15 nearest neighbors were considered to build the nearest neighbor graphs in all the tests. Each box shows the quartiles of the distribution, and the whiskers indicate the rest of the distribution except for the points that represent outliers. **c**, Bar plots of the ratio of cells that belong to aligned clusters between two example datasets where CAPITAL computed better than Seurat did ($\alpha = 0.9$ and $\beta = 0.6$). **d**, An opposite case of c with two different datasets ($\alpha = 0.9$ and $\beta = 0.6$).

Second, we have highlighted two examples where CAPITAL computed better than Seurat did and vice versa in Results section on pages 6–7 (Fig. 4c,d) along with Supplementary Figs. 1 and 2:

We further looked more carefully into a difference in alignment results between CAPITAL and Seurat. As one example, CAPITAL computed an alignment of two datasets better than Seurat did. Specifically, Seurat did not properly integrate the datasets, so that its alignment result was worse than that of CAPITAL (Fig. R5c). In contrast, Seurat calculated a good alignment of two other datasets that were successfully integrated, although CAPITAL yielded a passable result (Fig. R5d). These examples would suggest an advantage of CAPITAL over Seurat when aligning two trajectories in some cases.

Finally, one of the reasons why the data integration approach failed in some cases has been discussed in Discussion section on page 12:

Existing integration methods for matching cell clusters require the computation of anchors that link across different datasets to be compared, but such anchor-based data transformation still has a few limitations [6]. For example, there can be the wrong matching of cell subpopulations such as an integrated cluster with the imbalanced number of cells produced by Seurat as demonstrated in our test. On the other hand, for the same pair of the datasets, CAPITAL was able to compute as many matching clusters as possible. This will be attributed to the difference of dimensionality when aligning datasets: Seurat calculates mutual nearest neighbors in a low-dimensional space, whereas CAPITAL computes Spearman’s correlation in a high-dimensional expression space. Namely, the loss of information that is necessary to successful alignment might occur in the anchor-based integration in some cases.

Please note that “overcorrection” mentioned in the previous manuscript could be made in the non-linear batch correction algorithm [6], but this was not true to our case demonstrated in this paper. Thus, we have rewritten the paragraph as shown above.

- (2) *In Supp. Figure 1, I would expect CAPITAL would reach ABWAP score of around zero at very high noise levels. However, it seems like there is not so much of a decrease at all (maybe a very slow one). This tells me that the score might not be efficient to test the method’s accuracy. I suspect that this also affects the comparison between CAPITAL and Seurat where the first obtained only a marginal improvement at best. One possible solution would be creating one single cell data and then dividing it into different pairs. Different levels of noise can be added to*

each of the divided data. In that case, the true alignment is known so a measure of distance between the true and the predicted alignment can be assessed.

Response. We appreciate the reviewer’s helpful suggestion. We have performed a robustness test with newly generated datasets as suggested. The way of creating the new noise-added datasets is explained in Methods section on pages 20–21:

For benchmarking CAPITAL’s performance, each dataset was randomly split into two disjoint count matrices 100 times, resulting in $53 \times 100 = 5,300$ pairs of count matrices to be tested.

Gaussian noise was independently added 100 times to each of the 5,300 pairs of the log transformed count matrices, where the parameters of the distribution were set to zero mean and increasing standard deviation from 0 to 5.0 by step size 0.5. Note that the alignments of the 5,300 pairs without noise (i.e. zero mean and zero standard deviation) can be regarded as the true alignments, and thus comparison between the true alignment and the predicted alignment on the data with noise makes sense. To summarize, $5,300 \times 11 = 58,300$ pairs were used in the test for robustness.

Fig. R3. A performance indicator of CAPITAL as a function of the noise level on the synthetic datasets. The horizontal axis indicates the standard deviation of Gaussian noise that was independently added 100 times to each of the expression counts obtained by splitting a single-cell count matrix in the 53 synthetic datasets, which results in 58,300 pairs of trajectories. **a**, Performance evaluation in cluster–cluster alignment. Weight $\alpha = 0.9$ for the matched clusters to calculate the normalized alignment distance was used. **b**, Performance evaluation in cell–cell alignment. Penalty $\beta = 0.6$ for each unmatched cluster to compute the ABWAP score was used. 15 nearest neighbors were considered to build the nearest neighbor graphs in all the tests. Each box shows the quartiles of the distribution, and the whiskers indicate the rest of the distribution except for the points that represent outliers.

Using these datasets, we have tested the robustness of CAPITAL with both the

normalized alignment distance and the ABWAP score in Results section on pages 5–6 (Fig. 3):

Second, we tested the robustness of CAPITAL measured by alignment accuracy on the synthetic datasets with increasing data noise. More precisely, we evaluated the alignment accuracy from two measures: (i) the normalized alignment distance for assessing the performance of cluster–cluster alignment; and (ii) a metric defined on aligned trees based on area between worst and prediction (ABWAP) at a single-cell level. The rate of change in the normalized alignment distance was higher for the noise level of at least 3.5 than at most 3.0 (Fig. R3a), and that of the ABWAP score was higher for the noise level of at least 3.0 than at most 2.5 (Fig. R3b). Given that the noise of standard deviation 5.0 is unlikely to emanate from a typical dataset (e.g. standard deviation 5.0 was much larger than average 0.64 of the non-zero elements in the noise-free count matrices in our simulation), these results suggest that CAPITAL was robust to data noise to a certain degree at both a cluster-matching level and a single-cell alignment level.

For additional information on how big the noise of standard deviation 5.0 is, please see the following figure that shows PCA plots of the Leiden clustering of the same dataset to which Gaussian noise of standard deviation $\sigma \in \{1.0, 3.0, 5.0\}$ was added. These results indicate that clustering almost worked when adding noise of standard deviation 3.0, while shattered for noise of standard deviation 5.0.

Fig. R4. PCA plots of clustering results of a synthetic dataset. The number in parentheses that follows each graph title shows the standard deviation of Gaussian noise added to the data.

- (3) *In the gene enrichment analysis, it is not clear how and why genes are selected per cell type if the alignment is done per branch. Please elaborate on that.*

Response. We apologize for the ambiguity regarding the corresponding description in the previous manuscript. We did not select genes used in the analysis per cell type but per branch in the aligned tree. We have rephrased the misleading statements in Methods section on page 23. In addition, we would like to emphasize that those genes were computationally selected, not by prior knowledge of cell type.

References

- [1] Sakoe, H. & Chiba, S. Dynamic programming algorithm optimization for spoken word recognition. *IEEE T. Acoust. Speech* **26**, 43–49 (1978).
- [2] Alpert, A., Moore, L. S., Dubovik, T. & Shen-Orr, S. S. Alignment of single-cell trajectories to compare cellular expression dynamics. *Nat. Methods* **15**, 267–270 (2018).
- [3] Cacchiarelli, D. *et al.* Aligning single-cell developmental and reprogramming trajectories identifies molecular determinants of myogenic reprogramming outcome. *Cell Syst.* **7**, 258–268 (2018).
- [4] Wolf, F. A., Angerer, P. & Theis, F. J. SCANPY: large-scale single-cell gene expression data analysis. *Genome Biol.* **19**, 15 (2018).
- [5] Hao, Y. *et al.* Integrated analysis of multimodal single-cell data. *Cell* **184**, 3573–3587 (2021).
- [6] Argelaguet, R., Cuomo, A. S. E., Stegle, O. & Marioni, J. C. Computational principles and challenges in single-cell data integration. *Nat. Biotechnol.* **39**, 1202–1215 (2021).
- [7] Problem at reproducibility of UMAP / leiden. URL <https://github.com/theislab/scanpy/issues/1009>.

REVIEWER COMMENTS

Reviewer #2 (Remarks to the Author):

Major comments:

1. The authors show only marginal improvement over an integration framework using Seurat. While Seurat is a highly used method for single cell integration, other methods, such as scVI, tend to perform even better. Therefore, it is not clear, how minor advantages in distinct synthetic scenarios would be reflected on real life data.
2. While the authors changed their simulation framework, the ABWAP score is still ranging higher than zero even at very high noise levels. Indeed, even with noise level of 5 (which is considered to be high), the cell space in Figure R4 is still highly structured with distinct area for different cell types. It is important to adjust this analysis to show noise ranges from high success to total failure.

Suggestion: As of now, the work does not show any major advances over the current integration methods, like Seurat and others (and there are better approaches than Seurat). In addition, reference-quarry mapping tools, such as scArches, allows an automatic integration without even possessing the reference data. However, these methods rely on the proper integration as you showed in your small example (located in the Supp. Information). I feel like this is the main advantage of your tool. You do not need to show you are better than those methods in most cases, since you are probably not. However, you can find a niche where these methods fail in aligning the data and focus on that.

Reviewer #3 (Remarks to the Author):

The authors present CAPITAL, a novel algorithm for producing and aligning complex trajectories across scRNAseq datasets from different conditions. This seems like a very timely problem to address as scRNAseq is being leveraged in new ways to understand changes in complex systems. Despite the relevance, I think the authors have correctly identified a gap in the literature (namely, the alignment of complex, branching trajectories) and proposed a compelling solution.

I would also like to note that this is my first time reviewing this manuscript and I have only seen the authors' responses to the previous round of reviews, so I apologize for any redundancies or misunderstandings on my part. I will attempt to stay within the scope of the comments raised by the previous Reviewer #1, so as not to create unnecessary work for the authors.

MAJOR

(1)

The previous reviewer was skeptical about whether or not meaningful comparisons can be made between species without a baseline. I share this concern, however I don't see that as an issue for CAPITAL. I think that whether a particular change in expression occurs earlier or later in one species is a question that can only be answered in relative terms (ie. relative to other temporal markers). My only concern here is whether the dynamic time warping step uses only a single gene or multiple genes (as in cellAlign), allowing for these types of relative comparisons. My impression from the manuscript is that it only uses a single gene, but that would make such comparisons impossible, as pseudotime would be warped differently for each gene.

Also on the topic of dynamic time warping, the manuscript talks about mapping cells to cells. I know that the cellAlign algorithm uses interpolated points in local neighborhoods for added stability when performing dynamic time warping and I am wondering if CAPITAL does the same?

(2)

Assessment of clustering results. This is an important comment and the authors have clearly worked to address it, but I still have a few concerns.

First, the new metric "normalized alignment distance" (as well as ABWAP) seems inappropriate for measuring the performance of the Seurat integration pipeline, which is not based on the concept of "matched" and "unmatched" clusters. This forced the authors to create a new definition for whether single clusters were "matched" or "unmatched", namely that clusters must contain at least 25 cells from each condition to count as "matched". This definition is somewhat arbitrary (why 25? why not just 1?) and does not scale with cluster size (a small, balanced cluster with 24 cells from each condition would count as "unmatched"). This definition additionally seems to benefit CAPITAL in the comparison of methods, because it leads to more clusters being counted as "unmatched" in the Seurat analysis.

With all that said, I think there is simple solution: since the main comparison between CAPITAL and Seurat integration is being done on simulated data, wouldn't it make sense to use the true pseudotimes as a benchmark? Both methods ultimately produce pseudotime values (at least for a single lineage), so the correlation of these values with the known ground truth would seem like a simple metric for comparing them. You could also incorporate multiple lineages by averaging the correlation values for each lineage (possibly weighted by length or number of cells).

(3)

This comment seems to have been sufficiently addressed.

(4)

This comment seems to have been sufficiently addressed. I thought these results were quite interesting!

(5)

While the additional results are interesting, I unfortunately must maintain that the comparison between Seurat data integration and CAPITAL is not convincing. Primarily for the reasons detailed in (2), namely that the metrics for comparison are only appropriate to CAPITAL, not Seurat.

Additionally, I would point out that only about half of all simulated datasets were used and the process of selecting datasets was entirely based on the first two steps of the CAPITAL method. If these steps performed reasonably well, the dataset was retained, and if not, it wasn't. A more flexible metric for comparing methods (such as correlation with true pseudotime) could allow for a more transparent comparison by including some of these datasets on which (presumably) CAPITAL would have performed poorly.

(6)

This comment was sufficiently addressed. Minor point: in the dot plot figure legends, what does "Fraction of cells in group" mean? Does "group" mean "cluster"? Why is it almost always 100%?

(7)

The first part of this comment was addressed.

The authors claim that the "pseudotime arm" in the Setty et al. dataset is not a result of imputation, which may be true, but what is causing it?

(8)

This comment was addressed. I would note that there are other methods available for constructing cluster-based minimum spanning trees, some of which are able to account for cluster shape (such as distances based on mutual nearest neighbors). This may be a fruitful direction for future software

development.

(9)

Seems like a repeat of (1), though I agree that the phrase "relative temporal lag" is a bit unclear (relative to what?).

(10)

I had this question as well and it is interesting that CAPITAL did not perform as well for this comparison. That might be worth noting in the supplement.

(11)

I think this has been addressed, but I have a small follow-up. For the genes in Fig 6c, the authors claim that "most of them overlapped" with the computational results. It would be nice to be able to see this in the figure, such as by bolding the names of the genes that overlap.

(12)

This comment was addressed.

(13)

This comment was addressed.

(14)

This comment was addressed.

(15)

This comment was addressed.

(16)

This comment was addressed.

(17)

This comment was addressed (although I find the phrase "linear segments of single cells" to be very confusing. I think this could be replaced with "single lineages", but I'm not entirely sure what the authors mean by it).

MINOR

(1)

Figure 4 caption: "ans" -> "and"

(2)

For clarification: why does CAPITAL only calculate pseudotimes for a single lineage at a time? Diffusion pseudotime can be used to model branching trajectories and this would give additional stability (cells in the root node currently have different pseudotimes depending on which lineage is under consideration).

Responses to Reviewer #2's comments

Thank you very much for reading our revised manuscript many times and giving the helpful comments. We list our responses in roman type to the reviewer's comments italicized.

Please note that in this revision we have changed a few major/minor points that are summarized as follows:

- We have changed the synthetic datasets of scRNA-seq counts to more complex tree datasets with multiple branches (cf. the previous datasets contained only one branch in their trajectories, i.e. bifurcating structure), and increased the number of comparisons in the computational tests. Precisely, we have reanalyzed the tests on feasibility and robustness of CAPITAL, yet conclusions are the same. Moreover, we have compared alignment performance between CAPITAL and three data integration methods [1–3] on those datasets. We think that the new datasets will reflect real data more than previously presented (e.g. a population of hematopoietic cells tends to have multiple branches in the differentiation trajectory). These results clearly show an advantage of CAPITAL over the data integration methods in biological variance conservation.
- The metric for evaluating the performance of trajectory alignment at the single-cell level has been changed from the area between worst and prediction (ABWAP) [4] score to the average trajectory conservation (ATC) score [5] in order to measure biological variance conservation before and after alignment. The main reason is that the ABWAP score has parameter (penalty) β that is assigned to an unmatched pair of clusters in the CAPITAL framework. The problem of applying this metric to other integration methods lies in the fact that a threshold to decide whether a predicted cluster pair is matched or unmatched has to be determined. This means that the interpretation of the results strongly depends on the value of the threshold used, as the other reviewer pointed out. In contrast, the ATC score has no parameter and more straightforward.
- We have changed the way to select the starting cell of a linear trajectory in an aligned tree when computing diffusion pseudotime. Previously, the starting cell was defined as the cell that has a longest distance to a cell among others “along the linear trajectory,” while this time it has a longest distance to a cell among others “in the dataset.” The reason of this change is that it can work well even if the length of a linear trajectory is very short. Moreover, dynamic time warping has been performed with multiple genes instead of a single gene. Although some figures based on the pseudotime order have been changed, we confirmed that this does not affect our conclusion.

Major

- (1) *The authors show only marginal improvement over an integration framework using Seurat. While Seurat is a highly used method for single cell integration, other methods, such as scVI, tend to perform even better. Therefore, it is not clear, how minor advantages in distinct synthetic scenarios would be reflected on real life data.*

Response. We agree that there are other state-of-the-art tools for scRNA-seq data integration, and appreciate the reviewer’s suggestion. First, we have created the new datasets of complex binary trees (please see Methods section on page 21 in the revised manuscript). Second, we have newly defined the average trajectory conservation (ATC) score in Methods section on pages 20–21 as follows:

We define a parameter-free metric on an aligned trajectory tree of two scRNA-seq datasets to evaluate biological variation conservation before and after alignment, which is called an average trajectory conservation (ATC) score. Trajectory conservation was originally defined in the literature [5], and in this work we adapt the definition to our case where the tree structure should be explicitly considered to compute pseudotime. Precisely, for each single lineage that forms the aligned trajectory tree, a series of simulation times of the cells generated by dynngen [4] along that lineage is regarded as a reference, while a series of pseudotimes of aligned/integrated datasets as a prediction, between which Spearman’s rank correlation coefficient is calculated. Note that a starting cell of aligned/integrated datasets for all tools, which is necessary to compute diffusion pseudotime, is defined in a way that its simulation time in dynngen is 0 and it has the longest distance to a cell among all other cells in the expression space. In the case of CAPITAL, single-cell alignment is performed by dynamic time warping with two series of pseudotimes (e.g. for datasets 1 and 2), and thus the mean of the two respective correlation coefficients between the series of simulation times and either series of pseudotimes should be taken. Each Spearman’s rank correlation coefficient ρ computed above is scaled to the range $[0, 1]$ by $(\rho + 1)/2$, which we call a trajectory conservation score of a single lineage. Finally, the ATC score of the aligned trajectory tree is defined as the mean of the trajectory conservation scores of all single lineages. A high ATC score is considered to be an accurate alignment in trajectory conservation.

Third, we have run three data integration methods Scanorama [2], scVI [1] and Seurat [3], which is described in Methods section on page 22:

To compare the performance of other methods for integrating scRNA-

seq datasets, we used Scanorama 1.7.0 [2] and scVI 0.16.0 [1] as wrapper functions of single-cell integration benchmark (scib 1.0.3) [5], and Seurat 4.1.1 [3] to infer a common trajectory of each of all 2,278 pairs of the 68 synthetic datasets described above. More precisely, all pairs of the datasets in the form of either raw counts or preprocessed counts were integrated by the respective algorithms. A common trajectory was then estimated by finding a minimum spanning tree whose nodes were centroids as defined in the CAPITAL algorithm, which was considered as an aligned trajectory tree. Note that the same methods and the parameters as in CAPITAL were used to compute neighborhood graphs, Leiden clustering and diffusion pseudotime.

Lastly, the results of the comparative tests are described in Results section on pages 6–7:

Finally, to compare CAPITAL with three state-of-the-art methods of data integration in alignment performance, we ran Scanorama [2], scVI [1] and Seurat [3] on the combinations of the synthetic datasets to merge two respective datasets and perform common trajectory inference on that merged dataset. Note that pseudotime was computed for a trajectory tree of aligned and integrated datasets in CAPITAL and the other tools, respectively. We will show the superiority of one method over its competitors from two viewpoints: biological variance conservation and batch removal before and after alignment/integration [5]. First, the results measured by the ATC score as a metric of biological variance conservation indicate that CAPITAL was statistically significantly better than the data integration approaches (Fig. R1a). In particular, CAPITAL was more robust to the variation of the datasets that contained multiple branches than the other algorithms. Second, we demonstrate two examples of datasets on which CAPITAL achieved the most successful alignment, whereas the other algorithms failed to some degree or another (Fig. R1b,c,d and Supplementary Figs. 2 and 3). Specifically, CAPITAL was able to match all initial and terminal states, while Scanorama and scVI were unsuccessful in aligning some initial and terminal states, and Seurat partly failed to match initial states. In the end, CAPITAL achieved major advances over current integration methods in trajectory conservation for complex trajectory trees.

Fig. R1. Comparison of alignment accuracy between CAPITAL and data integration methods on a pair of synthetic datasets. **a**, Performance evaluation in trajectory conservation at the single-cell level on 2,278 pairs of the synthetic datasets. A one-sided Wilcoxon signed-rank test was performed for each pair of the methods. * * * indicates a p-value $< 1.00 \times 10^{-226}$, meaning that the ATC score of CAPITAL was significantly higher than those of the other tools. Each box shows the quartiles of the distribution, and the whiskers indicate the rest of the distribution except for the points that represent outliers. **b**, ATC scores of all tools on datasets 1 and 2. **c**, UMAP plots of datasets 1 and 2 with Leiden clustering, whose cell types were annotated by considering simulation time and expression patterns of transcription factors (Supplementary Fig. 2). The solid lines indicate the trajectories. The rightmost column shows an aligned trajectory tree of those datasets predicted by CAPITAL. “IS” and “TS” denote intermediate state and terminal state, respectively. **d**, UMAP plots of integration of datasets 1 and 2 computed by three data integration methods. The first and second columns indicate true simulation times in datasets 1 and 2, respectively, on the merged dataset, and the rightmost column shows UMAP plots of batch mixing. 10 nearest neighbors were considered to build the nearest neighbor graphs in all the tests.

Please note that we have not assessed batch removal by a quantitative metric such as an average silhouette width [5] but investigated UMAP plots qualitatively. One of the reason is that evaluation of batch removal requires an integrated data matrix, which is not calculated in CAPITAL. Second, the data integration methods are not based on the concept of “matched” and “unmatched” clusters in their integration results, which CAPITAL uses to evaluate its own performance by normalized alignment distance. Therefore, we have not compared batch removal quantitatively between CAPITAL and the other tools.

- (2) *While the authors changed their simulation framework, the ABWAP score is still ranging higher than zero even at very high noise levels. Indeed, even with noise level of 5 (which is considered to be high), the cell space in Figure R4 is still highly structured with distinct area for different cell types. It is important to adjust this analysis to show noise ranges from high success to total failure.*

Response. We apologize for the incomplete results without a high noise level that causes total failure. We have added noise whose standard deviation is beyond 5.0 of the previous maximum to 50.0, and confirmed that the noise level of 3.0 began to shatter the corresponding cell space for the new complex synthetic datasets. Please note that we have changed the metric from the ABWAP score to the ATC score as described above. Details of this noise test are described in Results section on pages 5–6:

Second, we tested the robustness of CAPITAL measured by alignment accuracy on the synthetic datasets with increasing data noise (Methods). More precisely, we evaluated the alignment accuracy from two measures: (i) the normalized alignment distance for assessing the performance of cluster–cluster alignment; and (ii) the average trajectory conservation (ATC) score at the single-cell level (Methods). The rate of change in the normalized alignment distance was higher for the noise level of at least 2.0 than at most 1.5 (Fig. R2a), and that in the ATC score was higher for the noise level of at least 1.5 than at most 1.0 (Fig. R2b). Asymptotically, the two metrics deteriorated for the noise level of 3.0 or higher, as the corresponding cell space began to be shattered (Supplementary Fig. 1). Note that the ATC score of around 0.5 means that a true simulation time and a predicted pseudotime are most likely to have no correlation. Given that the noise of standard deviation of around 3.0 is unlikely to emanate from a typical dataset (e.g. standard deviation 3.0 was much larger than average 0.53 of the non-zero elements in the noise-free count matrices in our simulation), these results suggest that CAPITAL was robust to data noise to a certain degree at both the cluster-matching level and the single-cell alignment level.

Fig. R2. A performance indicator of CAPITAL as a function of the noise level on the synthetic datasets. The horizontal axis indicates the standard deviation of Gaussian noise that was independently added 100 times to each of the expression counts obtained by splitting a single-cell count matrix in the 68 synthetic datasets, which results in 95,200 pairs of trajectories. **a**, Performance evaluation in cluster–cluster alignment. **b**, Performance evaluation in cell–cell alignment. 10 nearest neighbors were considered to build the nearest neighbor graphs in all the tests. Each box shows the quartiles of the distribution, and the whiskers indicate the rest of the distribution except for the points that represent outliers.

Suggestion

As of now, the work does not show any major advances over the current integration methods, like Seurat and others (and there are better approaches than Seurat). In addition, reference-query mapping tools, such as scArches, allows an automatic integration without even possessing the reference data. However, these methods rely on the proper integration as you showed in your small example (located in the Supp. Information). I feel like this is the main advantage of your tool. You do not need to show you are better than those methods in most cases, since you are probably not. However, you can find a niche where these methods fail in aligning the data and focus on that.

Response. We appreciate the helpful suggestion. By answering major point (1) to the best of our ability, we hope that the reviewer can check major advances of CAPITAL over the current integration methods in biological variance conservation and batch removal before and after alignment.

Responses to Reviewer #3's comments

Thank you very much for reading our revised manuscript carefully and giving the helpful comments. We list our responses in roman type to the reviewer's comments italicized.

Please note that in this revision we have changed a few major/minor points that are summarized as follows:

- We have changed the synthetic datasets of scRNA-seq counts to more complex tree datasets with multiple branches (cf. the previous datasets contained only one branch in their trajectories, i.e. bifurcating structure), and increased the number of comparisons in the computational tests. Precisely, we have reanalyzed the tests on feasibility and robustness of CAPITAL, yet conclusions are the same. Moreover, we have compared alignment performance between CAPITAL and three data integration methods [1–3] on those datasets. We think that the new datasets will reflect real data more than previously presented (e.g. a population of hematopoietic cells tends to have multiple branches in the differentiation trajectory). These results clearly show an advantage of CAPITAL over the data integration methods in biological variance conservation.
- The metric for evaluating the performance of trajectory alignment at the single-cell level has been changed from the area between worst and prediction (ABWAP) [4] score to the average trajectory conservation (ATC) score [5] in order to measure biological variance conservation before and after alignment. The main reason is that the ABWAP score has parameter (penalty) β that is assigned to an unmatched pair of clusters in the CAPITAL framework. The problem of applying this metric to other integration methods lies in the fact that a threshold to decide whether a predicted cluster pair is matched or unmatched has to be determined. This means that the interpretation of the results strongly depends on the value of the threshold used, as the reviewer pointed out. In contrast, the ATC score has no parameter and more straightforward.
- We have changed the way to select the starting cell of a linear trajectory in an aligned tree when computing diffusion pseudotime. Previously, the starting cell was defined as the cell that has a longest distance to a cell among others “along the linear trajectory,” while this time it has a longest distance to a cell among others “in the dataset.” The reason of this change is that it can work well even if the length of a linear trajectory is very short. Moreover, dynamic time warping has been performed with multiple genes instead of a single gene. Although some figures based on the pseudotime order have been changed, we confirmed that this does not affect our conclusion.

Summary

The authors present CAPITAL, a novel algorithm for producing and aligning complex trajectories across scRNAseq datasets from different conditions. This seems like a very timely problem to address as scRNAseq is being leveraged in new ways to understand changes in complex systems. Despite the relevance, I think the authors have correctly identified a gap in the literature (namely, the alignment of complex, branching trajectories) and proposed a compelling solution.

I would also like to note that this is my first time reviewing this manuscript and I have only seen the authors' responses to the previous round of reviews, so I apologize for any redundancies or misunderstandings on my part. I will attempt to stay within the scope of the comments raised by the previous Reviewer #1, so as not to create unnecessary work for the authors.

Response. We appreciate the reviewer's effort to make our study improved within the scope of the comments raised by the previous reviewer. Together with the other reviewer's suggestion, we have addressed all the points to the best of our ability, and believe that our paper has been much improved.

Major

- (1) *The previous reviewer was skeptical about whether or not meaningful comparisons can be made between species without a baseline. I share this concern, however I don't see that as an issue for CAPITAL. I think that whether a particular change in expression occurs earlier or later in one species is a question that can only be answered in relative terms (i.e. relative to other temporal markers). My only concern here is whether the dynamic time warping step uses only a single gene or multiple genes (as in cellAlign), allowing for these types of relative comparisons. My impression from the manuscript is that it only uses a single gene, but that would make such comparisons impossible, as pseudotime would be warped differently for each gene.*

Also on the topic of dynamic time warping, the manuscript talks about mapping cells to cells. I know that the cellAlign algorithm uses interpolated points in local neighborhoods for added stability when performing dynamic time warping and I am wondering if CAPITAL does the same?

Response. We appreciate the reviewer's understanding of our approach to comparisons between species without reference. To answer to the first question about use of multiple genes in dynamic time warping, we have already implemented the code that uses multiple genes so that relative comparisons can be made since the early version of CAPITAL. In the previous manuscript, however, we showed the results of dynamic time warping that used only a single gene as the reviewer

pointed out. We completely agree that pseudotime would be warped differently for each gene, and thus relative comparisons should be made using multiple genes. Therefore, we have reanalyzed and drawn Figs. 5d,e and 6c,d, and Supplementary Figs. 8, 12 and 13 that were obtained from the processes with dynamic time warping. Please note again that this change does not affect our conclusion.

As for the second question, we think that we can incorporate interpolated points into the phase of cell–cell alignment in CAPITAL, as done in cellAlign. We agree that such extension would be of interest to have more stable results. However, we feel that interpolated points are artifacts even though it would help add the stability, and we have already performed cluster–cluster alignment to resolve difference to cell–cell matching to some degree.

- (2) *Assessment of clustering results. This is an important comment and the authors have clearly worked to address it, but I still have a few concerns.*

First, the new metric “normalized alignment distance” (as well as ABWAP) seems inappropriate for measuring the performance of the Seurat integration pipeline, which is not based on the concept of “matched” and “unmatched” clusters. This forced the authors to create a new definition for whether single clusters were “matched” or “unmatched,” namely that clusters must contain at least 25 cells from each condition to count as “matched.” This definition is somewhat arbitrary (why 25? why not just 1?) and does not scale with cluster size (a small, balanced cluster with 24 cells from each condition would count as “unmatched”). This definition additionally seems to benefit CAPITAL in the comparison of methods, because it leads to more clusters being counted as “unmatched” in the Seurat analysis.

With all that said, I think there is simple solution: since the main comparison between CAPITAL and Seurat integration is being done on simulated data, wouldn't it make sense to use the true pseudotimes as a benchmark? Both methods ultimately produce pseudotime values (at least for a single lineage), so the correlation of these values with the known ground truth would seem like a simple metric for comparing them. You could also incorporate multiple lineages by averaging the correlation values for each lineage (possibly weighted by length or number of cells).

Response. We completely agree with the reviewer's concerns and appreciate the helpful suggestion. As described at the top of our responses, we have newly defined the parameter-free score, which is described in Methods section on pages 20–21:

We define a parameter-free metric on an aligned trajectory tree of two scRNA-seq datasets to evaluate biological variation conservation before and after alignment, which is called an average trajectory conservation

(ATC) score. Trajectory conservation was originally defined in the literature [5], and in this work we adapt the definition to our case where the tree structure should be explicitly considered to compute pseudotime. Precisely, for each single lineage that forms the aligned trajectory tree, a series of simulation times of the cells generated by *dynngen* [4] along that lineage is regarded as a reference, while a series of pseudotimes of aligned/integrated datasets as a prediction, between which Spearman's rank correlation coefficient is calculated. Note that a starting cell of aligned/integrated datasets for all tools, which is necessary to compute diffusion pseudotime, is defined in a way that its simulation time in *dynngen* is 0 and it has the longest distance to a cell among all other cells in the expression space. In the case of CAPITAL, single-cell alignment is performed by dynamic time warping with two series of pseudotimes (e.g. for datasets 1 and 2), and thus the mean of the two respective correlation coefficients between the series of simulation times and either series of pseudotimes should be taken. Each Spearman's rank correlation coefficient ρ computed above is scaled to the range $[0, 1]$ by $(\rho + 1)/2$, which we call a trajectory conservation score of a single lineage. Finally, the ATC score of the aligned trajectory tree is defined as the mean of the trajectory conservation scores of all single lineages. A high ATC score is considered to be an accurate alignment in trajectory conservation.

Please note that the final average has not been weighted by the number of cells because we have computed correlation coefficients, which would be independent of the number of cells.

Next, we have run three data integration methods Scanorama [2], scVI [1] and Seurat [3], which is described in Methods section on page 22:

To compare the performance of other methods for integrating scRNA-seq datasets, we used Scanorama 1.7.0 [2] and scVI 0.16.0 [1] as wrapper functions of single-cell integration benchmark (scib 1.0.3) [5], and Seurat 4.1.1 [3] to infer a common trajectory of each of all 2,278 pairs of the 68 synthetic datasets described above. More precisely, all pairs of the datasets in the form of either raw counts or preprocessed counts were integrated by the respective algorithms. A common trajectory was then estimated by finding a minimum spanning tree whose nodes were centroids as defined in the CAPITAL algorithm, which was considered as an aligned trajectory tree. Note that the same methods and the parameters as in CAPITAL were used to compute neighborhood graphs, Leiden clustering and diffusion pseudotime.

Lastly, the results of the comparative tests are described in Results section on

pages 6–7:

Finally, to compare CAPITAL with three state-of-the-art methods of data integration in alignment performance, we ran Scanorama [2], scVI [1] and Seurat [3] on the combinations of the synthetic datasets to merge two respective datasets and perform common trajectory inference on that merged dataset. Note that pseudotime was computed for a trajectory tree of aligned and integrated datasets in CAPITAL and the other tools, respectively. We will show the superiority of one method over its competitors from two viewpoints: biological variance conservation and batch removal before and after alignment/integration [5]. First, the results measured by the ATC score as a metric of biological variance conservation indicate that CAPITAL was statistically significantly better than the data integration approaches (Fig. R1a). In particular, CAPITAL was more robust to the variation of the datasets that contained multiple branches than the other algorithms. Second, we demonstrate two examples of datasets on which CAPITAL achieved the most successful alignment, whereas the other algorithms failed to some degree or another (Fig. R1b,c,d and Supplementary Figs. 2 and 3). Specifically, CAPITAL was able to match all initial and terminal states, while Scanorama and scVI were unsuccessful in aligning some initial and terminal states, and Seurat partly failed to match initial states. In the end, CAPITAL achieved major advances over current integration methods in trajectory conservation for complex trajectory trees.

Fig. R1. Comparison of alignment accuracy between CAPITAL and data integration methods on a pair of synthetic datasets. **a**, Performance evaluation in trajectory conservation at the single-cell level on 2,278 pairs of the synthetic datasets. A one-sided Wilcoxon signed-rank test was performed for each pair of the methods. * * * indicates a p-value $< 1.00 \times 10^{-226}$, meaning that the ATC score of CAPITAL was significantly higher than those of the other tools. Each box shows the quartiles of the distribution, and the whiskers indicate the rest of the distribution except for the points that represent outliers. **b**, ATC scores of all tools on datasets 1 and 2. **c**, UMAP plots of datasets 1 and 2 with Leiden clustering, whose cell types were annotated by considering simulation time and expression patterns of transcription factors (Supplementary Fig. 2). The solid lines indicate the trajectories. The rightmost column shows an aligned trajectory tree of those datasets predicted by CAPITAL. “IS” and “TS” denote intermediate state and terminal state, respectively. **d**, UMAP plots of integration of datasets 1 and 2 computed by three data integration methods. The first and second columns indicate true simulation times in datasets 1 and 2, respectively, on the merged dataset, and the rightmost column shows UMAP plots of batch mixing. 10 nearest neighbors were considered to build the nearest neighbor graphs in all the tests.

Please note that we have not assessed batch removal by a quantitative metric such as an average silhouette width [5] but investigated UMAP plots qualitatively. One of the reason is that evaluation of batch removal requires an integrated data matrix, which is not calculated in CAPITAL. Second, the data integration methods are not based on the concept of “matched” and “unmatched” clusters in their integration results, which CAPITAL uses to evaluate its own performance by normalized alignment distance. Therefore, we have not compared batch removal quantitatively between CAPITAL and the other tools.

- (3) *This comment seems to have been sufficiently addressed.*
- (4) *This comment seems to have been sufficiently addressed. I thought these results were quite interesting!*

Response. As the other reviewer pointed out that noise used in the test should range from high success to total failure, we have reanalyzed the robustness test as described in Results section on pages 5–6:

Second, we tested the robustness of CAPITAL measured by alignment accuracy on the synthetic datasets with increasing data noise (Methods). More precisely, we evaluated the alignment accuracy from two measures: (i) the normalized alignment distance for assessing the performance of cluster–cluster alignment; and (ii) the average trajectory conservation (ATC) score at the single-cell level (Methods). The rate of change in the normalized alignment distance was higher for the noise level of at least 2.0 than at most 1.5 (Fig. R2a), and that in the ATC score was higher for the noise level of at least 1.5 than at most 1.0 (Fig. R2b). Asymptotically, the two metrics deteriorated for the noise level of 3.0 or higher, as the corresponding cell space began to be shattered (Supplementary Fig. 1). Note that the ATC score of around 0.5 means that a true simulation time and a predicted pseudotime are most likely to have no correlation. Given that the noise of standard deviation of around 3.0 is unlikely to emanate from a typical dataset (e.g. standard deviation 3.0 was much larger than average 0.53 of the non-zero elements in the noise-free count matrices in our simulation), these results suggest that CAPITAL was robust to data noise to a certain degree at both the cluster-matching level and the single-cell alignment level.

- (5) *While the additional results are interesting, I unfortunately must maintain that the comparison between Seurat data integration and CAPITAL is not convincing. Primarily for the reasons detailed in (2), namely that the metrics for comparison are only appropriate to CAPITAL, not Seurat.*

Fig. R2. A performance indicator of CAPITAL as a function of the noise level on the synthetic datasets. The horizontal axis indicates the standard deviation of Gaussian noise that was independently added 100 times to each of the expression counts obtained by splitting a single-cell count matrix in the 68 synthetic datasets, which results in 95,200 pairs of trajectories. **a**, Performance evaluation in cluster–cluster alignment. **b**, Performance evaluation in cell–cell alignment. 10 nearest neighbors were considered to build the nearest neighbor graphs in all the tests. Each box shows the quartiles of the distribution, and the whiskers indicate the rest of the distribution except for the points that represent outliers.

Additionally, I would point out that only about half of all simulated datasets were used and the process of selecting datasets was entirely based on the first two steps of the CAPITAL method. If these steps performed reasonably well, the dataset was retained, and if not, it wasn't. A more flexible metric for comparing methods (such as correlation with true pseudotime) could allow for a more transparent comparison by including some of these datasets on which (presumably) CAPITAL would have performed poorly.

Response. We think that our response to major comment (2) can answer to the first point.

As for the second point, we have used the new datasets such that the number of leaves in a minimum spanning tree based on Leiden clustering is exactly four. We think that this criterion is more general than previously used. Indeed, the data selection was done before alignment and might benefit CAPITAL, but guarantees the structure of the datasets generated by dyngen [4], which specified binary trees with three branches in the simulation. This can also help the other tools build suitable trajectory trees after integration.

(6) *This comment was sufficiently addressed.*

Minor point: in the dot plot figure legends, what does “Fraction of cells in group” mean? Does “group” mean “cluster”? Why is it almost always 100%?

Response. As the reviewer guessed, “fraction of cells in group” means fraction

of cells in cluster. This notation has been automatically shown in Scanpy [6], which we have used to draw the figures. One possible reason why it is almost always 100% in Supplementary Figs. 5a, 7a and 11a is that the genes appeared in columns are highly variable genes across all cells.

- (7) *The first part of this comment was addressed.*

The authors claim that the “pseudotime arm” in the Setty et al. dataset is not a result of imputation, which may be true, but what is causing it?

Response. We speculate that the pseudotime arm might be caused by technical issues when experimentally capturing mRNAs that cannot be recovered by imputation, or by the unsuitable pseudotime setting. Please note that we have changed Fig. 5d,e from GATA1 along the erythrocyte lineage to ITGA2B along the megakaryocyte lineage to show a clear message to readers. Specifically, all kinetics curves have been slightly changed due to the use of multiple genes during dynamic time warping, and the pseudotime arm in GATA1 figure has been absorbed (Supplementary Fig. 8).

- (8) *This comment was addressed.*

I would note that there are other methods available for constructing cluster-based minimum spanning trees, some of which are able to account for cluster shape (such as distances based on mutual nearest neighbors). This may be a fruitful direction for future software development.

Response. We appreciate the future direction that the reviewer noted. We hope to consider the ability to explain cluster shape to our method in a future study.

- (9) *Seems like a repeat of (1), though I agree that the phrase “relative temporal lag” is a bit unclear (relative to what?).*

Response. We meant relative temporal lag to the other dataset. We have removed the corresponding sentence to avoid confusion.

- (10) *I had this question as well and it is interesting that CAPITAL did not perform as well for this comparison. That might be worth noting in the supplement.*

Response. We apologize for any inconvenience that hinders deep understanding of the cross-species trajectory alignment. We have added Supplementary Fig. 9 to explain that CAPITAL did not perform as well on the pair of Setty et al.’s data and Paul et al.’s data.

- (11) *I think this has been addressed, but I have a small follow-up. For the genes in Fig 6c, the authors claim that “most of them overlapped” with the computational results. It would be nice to be able to see this in the figure, such as by bolding the names of the genes that overlap.*

Response. We have added an asterisk to a gene name that was NOT overlapped with the results of the computational screen in Fig. 6c.

(12) *This comment was addressed.*

(13) *This comment was addressed.*

(14) *This comment was addressed.*

(15) *This comment was addressed.*

(16) *This comment was addressed.*

(17) *This comment was addressed (although I find the phrase “linear segments of single cells” to be very confusing. I think this could be replaced with “single lineages”, but I’m not entirely sure what the authors mean by it).*

Response. We have replaced the confusing phrase with “single lineages” as suggested.

Minor

(1) *Figure 4 caption: “ans” → “and”*

Response. We have corrected the typo.

(2) *For clarification: why does CAPITAL only calculate pseudotimes for a single lineage at a time? Diffusion pseudotime can be used to model branching trajectories and this would give additional stability (cells in the root node currently have different pseudotimes depending on which lineage is under consideration).*

Response. We agree that use of diffusion pseudotime for all lineages would be of interest. However, we feel it rather unstable from the following reasons. First, branches in a minimum spanning tree in CAPITAL’s framework may not be compatible with branches that the diffusion pseudotime algorithm detects. This would be true for other algorithms such as Palantir [7]. Second, we have found in Scanpy’s API [8] that diffusion pseudotime is not recommended to calculate pseudotimes of all trajectories.

References

- [1] Lopez, R., Regier, J., Cole, M. B., Jordan, M. I. & Yosef, N. Deep generative modeling for single-cell transcriptomics. *Nat. Methods* **15**, 1053–1058 (2018).
- [2] Hie, B., Bryson, B. & Berger, B. Efficient integration of heterogeneous single-cell transcriptomes using Scanorama. *Nat. Biotechnol.* **37**, 685–691 (2019).
- [3] Hao, Y. *et al.* Integrated analysis of multimodal single-cell data. *Cell* **184**, 3573–3587 (2021).
- [4] Cannoodt, R., Saelens, W., Deconinck, L. & Saeys, Y. Spearheading future omics analyses using dyngen, a multi-modal simulator of single cells. *Nat. Commun.* **12**, 3942 (2021).
- [5] Luecken, M. D. *et al.* Benchmarking atlas-level data integration in single-cell genomics. *Nat. Methods* **19**, 41–50 (2022).
- [6] Wolf, F. A., Angerer, P. & Theis, F. J. SCANPY: large-scale single-cell gene expression data analysis. *Genome Biol.* **19**, 15 (2018).
- [7] Setty, M. *et al.* Characterization of cell fate probabilities in single-cell data with Palantir. *Nat. Biotechnol.* **37**, 451–460 (2019).
- [8] scanpy.tl.dpt. URL <https://scanpy.readthedocs.io/en/latest/generated/scanpy.tl.dpt.html#scanpy.tl.dpt>.

REVIEWERS' COMMENTS

Reviewer #3 (Remarks to the Author):

Hello,

Once again, I am taking over for another reviewer and will attempt to continue their line of thought. Thankfully, they only left two comments this time, rather than seventeen.

I think comment (1) and the closing (Suggestion) are highly interconnected, so I will combine them.

(1) I think this comment has been partially addressed. The addition of scVI and Scanorama to the simulation study definitely strengthens that section and I think the original reviewer *should* be satisfied with that (in the Suggestion, they propose including scArches, which isn't exactly comparable and was published earlier this year, so if that's fair game, then one could continue asking for more comparisons indefinitely).

However, there is another point in these comments that I think the authors have not addressed, which is the advantage (if any) of using CAPITAL on *real data*. Both of the real-world analyses presented in the paper use CAPITAL and show that meaningful biological results are obtained. But neither explores the question of whether or not those same results could have been obtained by using Seurat/scVI. The reviewer encouraged the authors to "find a niche where these methods fail" and I agree that such a result is currently missing and would strengthen the paper.

If this were a clustering or trajectory inference method, then I think this sort of real-world evidence would be critically important. As it is, aligning trajectories is a fairly novel concept, as evidenced by the fact that none of the existing methods were specifically designed for that purpose.

(2) This comment has been thoroughly addressed.

Responses to Reviewer #3's comments

Thank you very much for reading our revised manuscript and giving the helpful comments. We list our responses in roman type to the reviewer's comments italicized.

- (1) *I think this comment has been partially addressed. The addition of scVI and Scanorama to the simulation study definitely strengthens that section and I think the original reviewer *should* be satisfied with that (in the Suggestion, they propose including scArches, which isn't exactly comparable and was published earlier this year, so if that's fair game, then one could continue asking for more comparisons indefinitely).*

*However, there is another point in these comments that I think the authors have not addressed, which is the advantage (if any) of using CAPITAL on *real data*. Both of the real-world analyses presented in the paper use CAPITAL and show that meaningful biological results are obtained. But neither explores the question of whether or not those same results could have been obtained by using Seurat/scVI. The reviewer encouraged the authors to "find a niche where these methods fail" and I agree that such a result is currently missing and would strengthen the paper.*

If this were a clustering or trajectory inference method, then I think this sort of real-world evidence would be critically important. As it is, aligning trajectories is a fairly novel concept, as evidenced by the fact that none of the existing methods were specifically designed for that purpose.

Response. We appreciate the reviewer's careful reading of our revised paper and the helpful comment. We agree that such real-data analysis would be of interest. However, we do not think that additional work is necessary to further strengthen the present research by the following reasons:

- The simulation datasets generated by dyngen are based on real reference datasets and have multiple branches in our tests, and thus we think that they reflect real data such as hematopoietic cells with multi-branching potential to a certain degree as mentioned in our previous response. Although the number of cells in the simulation tends to be smaller than in the real settings, we think that much higher number of synthetic datasets used in the benchmarking will complement results of real biological data. In our opinion, it is normal to find out a best tool in benchmarking and use it in the subsequent real-world analyses, as real datasets do not have shared ground truth to validate trajectory alignments, and meaningful comparison between different methods would be difficult.
- Our goal is to provide a new method for "aligning" branched trajectories, not for inferring those, as the reviewer noticed. Although CAPITAL is also capable

to estimate branched trajectories by itself, it can accept trajectories precomputed by a third-party tool, which was already described in Discussion section on page 9.

A summary of the first point described above has been added to Methods section on page 15 in the final revised manuscript.

(2) *This comment has been thoroughly addressed.*

Response. We thank for the confirmation.